# Cellular and subcellular heterogeneity of astrocytic Na⁺ homeostasis tuning astrocytes into functionally distinct subgroups in the mouse brain

Jan Meyer [1], Viola Bornemann [1,6], Alok Bhattarai [2,6], Sara Eitelmann[1], Petr Unichenko[3], Simone Durry[1], Karl W. Kafitz[1], Nicholas Chalmers [4], Jianfeng Fan [1], Ruth Beckervordersandforth [4], Christian Henneberger [3,5], Ghanim Ullah [2] & Christine R. Rose [1] ✉

Astrocytes maintain extracellular ion and transmitter homeostasis, with the Na⁺ inward gradient playing a crucial role. Earlier studies suggested a rather low, uniform Na⁺ distribution in astrocytes, consistent with the view that these basic homeostatic properties are well-protected. Here, we employed multi-photon fluorescence lifetime imaging to quantitatively determine astrocytic [Na⁺] in mouse brain tissue slices and in vivo. Our data reveals a significant subcellular and cellular heterogeneity in astrocytic [Na⁺], accompanied by differences in the capacity for Na⁺/K⁺-ATPase (NKA)-mediated uptake of extracellular K⁺. RNAscope and immunohistochemistry indicate differential spatial expression patterns of NKA ß1 and ß2 subunits in astrocytes. Biophysical modeling of differential NKA expression together with varying strength of Na⁺ influx replicate the experimentally observed heterogeneity in astrocytic [Na⁺]. Altogether, our results suggest the existence of functionally distinct astrocytes and astrocyte subdomains in which Na⁺ homeostasis is locally adapted to the specific requirements of surrounding neural networks.

Astrocytes are essential for the proper functioning of the vertebrate brain. They contribute to the formation and plasticity of neural networks[1], and are critically involved in the ionic homeostasis of the extracellular space (ECS)[2,3]. This includes control of extracellular K⁺ ($[K^+]_e$) through uptake of K⁺ via Kir4.1 channels and the Na⁺/K⁺-ATPase (NKA)[4,5], by which astrocytes control neuronal excitability and network performance[6]. In addition, the NKA represents the dominant mechanism for Na⁺ export, establishing a low intracellular Na⁺ concentration ($[Na^+]$) against a high inward gradient. The latter provides the driving force for a multitude of transporters, including Na⁺-dependent glutamate transporters (excitatory amino acid transporters, EAATs), making a low $[Na^+]$ and NKA activity the core and hub of astrocyte function[7,8]. Importantly, the maintenance of a low $[Na^+]$ is constantly challenged by local or global Na⁺ influx, e. g., upon synaptic release of glutamate and its uptake by astrocytes, counteracted by NKA-mediated Na⁺ efflux upon increases in $[K^+]_e$[7,9-11].

The mean values determined for astrocyte $[Na^+]$ in rodent tissue slices are around 12–15 mM[7,9,10]. This data, however, represents bulk measurements from cell bodies, and it is currently unclear whether somatic $[Na^+]$ ($[Na^+]_s$) also reflects $[Na^+]$ in glial processes ($[Na^+]_p$). Due

[1]Institute of Neurobiology, Faculty of Mathematics and Natural Sciences, Heinrich Heine University Düsseldorf, Düsseldorf, Germany. [2]Department of Physics, University of South Florida, Tampa, FL, USA. [3]Institute of Cellular Neurosciences I, Medical Faculty, University of Bonn, Bonn, Germany. [4]Department of Neurosurgery, University Hospital Erlangen, Friedrich-Alexander-Universität, Erlangen, Germany. [5]German Center for Neurodegenerative Diseases, Bonn, Germany. [6]These authors contributed equally: Viola Bornemann, Alok Bhattarai. ✉e-mail: rose@hhu.de

to the high mobility of $Na^+$ and the absence of relevant $Na^+$-buffers, it is commonly assumed that at rest, $[Na^+]$ is largely equilibrated within individual astrocytes and between gap-junction-coupled syncytia[12,13]. Since a low $[Na^+]$ is crucial for ionic and transmitter homeostasis, one could suspect that baseline $[Na^+]$ is mostly identical in different astrocytes and astrocyte sub-compartments to ensure the reliable fulfillment of these vital astrocytic functions. Recent modeling work, however, challenged this concept, proposing the existence of microdomains and subcellular gradients for $Na^{+}$[14,15]. Along this line, large local depolarizations mediated by neuronal activity and activation of EAATs were reported from peripheral astrocyte processes, indicating local ($Na^+$-) signaling domains[16]. Moreover, the efficacy of local glutamate uptake by astrocytes depends on the spine size[17] as well as on the spatial proximity of an astrocyte process to a synapse[18], pointing towards a likely subcellular heterogeneity of $Na^+$ influx.

As mentioned above, there is currently no experimental data addressing potential heterogeneities in astrocytic $[Na^+]$. One reason is that the unbiased determination of astrocytic $[Na^+]$ is challenging. At present, there are no genetically encoded $Na^+$ indicators suitable for experiments in intact brain tissue. Moreover, intensity-based imaging using chemical indicators is prone to artefacts related to differences in, or changes of, dye concentrations. An alternative approach is fluorescence lifetime imaging microscopy (FLIM), which is essentially independent of dye concentrations[19], and has for example, enabled quantitative analysis of astrocytic $Ca^{2+}$ or $Cl^-$ concentrations[5,20–23].

Here, we established multi-photon FLIM (MP-FLIM) based on time-correlated single-photon counting (TCSPC) of the $Na^+$ indicator ION Natrium Green-2 (ING-2)[24,25] for quantitative, dynamic measurement of astrocytic $[Na^+]$ in mouse forebrain tissue slices and in vivo. Overall, our results reveal a substantial cellular and subcellular heterogeneity in astrocytic $Na^+$-homeostasis, resulting in functionally distinct astrocyte subgroups and subdomains, optimized for efficient regulation of homeostasis in the ECS.

## Results

### MP-FLIM-based measurement of astrocytic $[Na^+]_s$ in situ and in vivo

To establish MP-FLIM of astrocytic $[Na^+]_s$, hippocampal slices were bolus-loaded with the membrane-permeable form of ING-2 (Fig. 1a), resulting in the staining of neurons and SR101-labeled astrocytes in the CA1 area. For recording of the fluorescence lifetime (FL) of ING-2, regions of interest (ROIs) were placed around somata of neurons in the pyramidal cell layer and of SR101-positive astrocytes in the *stratum radiatum* (Fig. 1b). Slices were then perfused with different calibration salines containing gramicidin ($Na^+$ ionophore), monensin ($Na^+/H^+$ antiporter), ouabain (NKA inhibitor) and a defined $[Na^+]$ (0–150 mM) to equilibrate extra- and intracellular $[Na^+]$ (Fig. 1c, d). The change in ING-2 FL with increasing $[Na^+]_s$ followed a Michaelis-Menten relationship with an apparent $K_D$ of 21 mM and was virtually identical in neuronal and astrocyte somata as well as in ROIs drawn around astrocyte processes (Fig. 1e), demonstrating that MP-FLIM of ING-2 is well suited for the determination of $[Na^+]$ in these different compartments.

To establish baseline astrocytic $[Na^+]_s$, the ING-2 FL was determined in somata of astrocytes perfused with standard saline, revealing a mean $[Na^+]_s$ of $13.8 \pm 7.2$ mM ($n = 369$, $N = 32$) (Fig. 1f). The vast majority of cells (80%) exhibited a $[Na^+]_s$ between 5 and 20 mM (total range: 2–46 mM). A mixed linear regression analysis revealed no relationship between $[Na^+]_s$ of astrocytes and the number of slice preparations or mice used ($p = 0.586$). Furthermore, $[Na^+]_s$ was independent of the relative depth, and cells with high and low baseline $[Na^+]_s$ were found close to the slice surface as well as in deeper tissue layers (Pearson: 0.17, $R^2 = 0.03$) (Fig. 1g). The overall distribution of $[Na^+]_s$ could be best described by a double Gaussian fit ($R^2 = 0.97$), centered around $8.7 \pm 0.5$ mM and $17.0 \pm 0.6$ mM (Fig. 1h).

To validate $[Na^+]_s$ as reported by FLIM and the respective calibrations, ING-2 FL of astrocytes was first measured in standard saline, resulting in a mean $[Na^+]_s$ of $12.8 \pm 6.4$ mM ($n = 13$, $N = 4$) (Fig. 1i, j). Afterwards, slices were perfused with $Na^+$-free calibration saline. Independent of their former baseline, washout of $Na^+$ caused the FL in somata to drop to values corresponding near 0 mM $[Na^+]_s$ (mean: $1.3 \pm 1.5$ mM) within about 2 minutes (Fig. 1i, j). In another set of experiments, slices were perfused with $Na^+$-free saline. This resulted in a washout of intracellular $Na^+$ to values close to 0 mM (mean: $0.5 \pm 0.8$ mM) within 15–20 min. Switching to $Na^+$-free calibration saline did not further change $[Na^+]_s$ (mean: $0.4 \pm 0.7$ mM, $p = 0.26$) (Fig. 1j). Finally, we performed widefield imaging with the ratiometric $Na^+$ indicator SBFI-AM[26]. Converting the somatic SBFI-ratio values based on calibrations analogous to those for ING-2 FL, revealed an average $[Na^+]_s$ of $13.1 \pm 5.4$ mM ($n = 46$, $N = 8$) (Suppl. Figure 1), which is similar to that determined by MP-FLIM of ING-2 ($p = 0.85$). As before, $[Na^+]_s$ exhibited a large range (4.9-24.9 mM) and was best fit by the same bimodal distribution (unimodal: $R^2 = 0.43$; bimodal: $R^2 = 0.64$) (Supplementary Fig. 1). These results confirm that ING-2 FL reliably reports astrocytic $[Na^+]$.

Astrocyte properties differ among brain regions[27–30]. To reveal potential differences between hippocampal and cortical astrocytes, we performed MP-FLIM of ING-2 in layers 2/3 of cortical slices (Fig. 2a–d). Average $[Na^+]_s$ of cortical astrocytes was similar to those of hippocampal astrocytes (mean: $12.5 \pm 4.2$ mM, range: 6.3-29.6 mM; $n = 34$, $N = 4$; $p = 0.53$) (Fig. 2c). As observed before, $[Na^+]_s$ was independent of the tissue depth (Pearson: 0.04, $R^2 = 0.002$) (Fig. 2d).

Finally, we also assessed $[Na^+]_s$ in vivo, performing MP-FLIM of ING-2 in SR101-positive cells of cortical layers 2/3 of anesthetized mice (Fig. 2e–g and Supplementary Fig. 2). A reliable intracellular calibration as above cannot be performed in vivo. Therefore, we first employed the in situ calibration parameters and apparent in situ $K_D$ (see above), which resulted in nonsensical values (< 0 mM) in 40% of astrocytes. This is plausible because of the different optics and recording conditions. We then used a separate setup-specific calibration in this set of experiments (see "Methods"). The resulting estimated average astrocytic $[Na^+]_s$ was $19.8 \pm 10.6$ mM (range: 3.7–42.1 mM; $n = 50$, 12 field of views) (Fig. 2g). This estimated value for $[Na^+]_s$ in vivo was significantly higher than $[Na^+]_s$ determined in cortical ($p = 0.002$) and hippocampal ($p = 0.00017$) slices.

These results demonstrate that MP-FLIM of ING-2 enables determination of $[Na^+]_s$ of astrocytes in situ and in vivo. They confirm earlier work showing that the mean $[Na^+]_s$ of hippocampal and cortical astrocytes is ~13 mM in tissue slice preparations[7]. However, they also reveal that $[Na^+]_s$ exhibits a large intercellular heterogeneity and displays a bimodal distribution. In addition, our data suggests that baseline $[Na^+]_s$ is increased by several mM in cortical astrocytes in vivo.

### $[Na^+]$ in astrocyte processes

For quantification of $[Na^+]$ in processes ($[Na^+]_p$), individual SR101-positive cells were loaded with ING-2 using whole-cell patch-clamp (Fig. 3a). Cells exhibited a resting membrane potential of $-84.8 \pm 0.4$ mV and a linear I/V relationship, typical for mature astrocytes (n/N = 24) (Fig. 3b). Z-stacks of ING-2 FL were taken, covering a depth of at least 50 μm (Fig. 3a). Somata of cells held in whole-cell were excluded from analysis (Fig. 3a), because somata are quickly dialyzed by the intracellular saline, preventing a meaningful conversion of ING-2 FL into $[Na^+]$[25]. ROIs were drawn around individual processes arising from the soma, including clearly visible branchpoints and averaged for a given cell (Fig. 3a), resulting in a mean baseline $[Na^+]_p$ of $17.3 \pm 6.5$ mM (range: 5–27 mM; n/N = 29). This value is significantly higher than that of somata in bolus-loaded slices ($p = 0.004$) (Fig. 3b). Of note, analysis of FL was restricted to processes exhibiting > 5 photons/pixel/frame to enable reliable fitting of photon distributions (Supplementary Fig. 3).

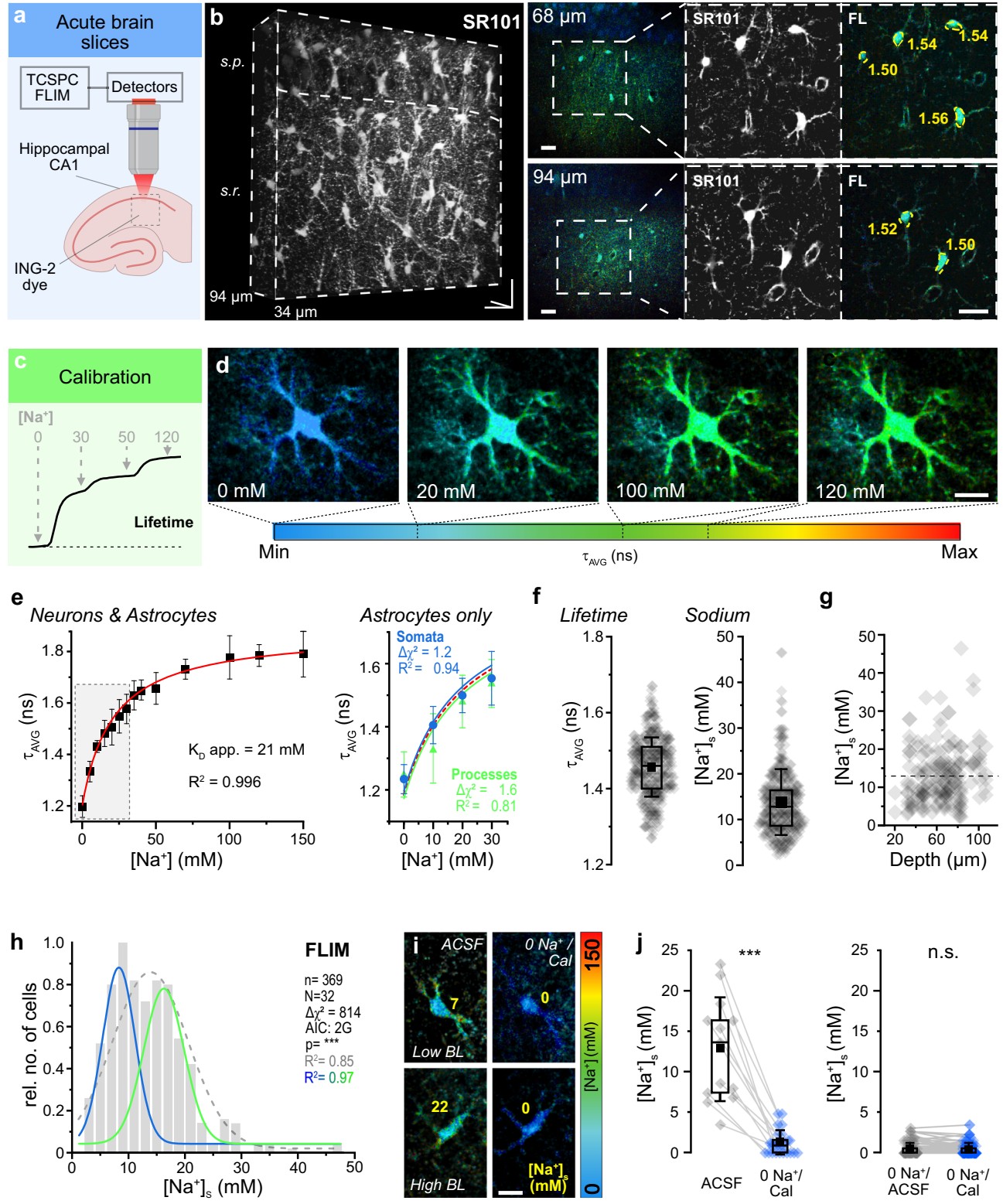

Summing up photons by increasing the binning from 1 to up to 10 resulted in virtually identical FL, demonstrating that our minimum criterium was sufficient for proper determination of $[Na^+]_p$ (Supplementary Fig. 3).

Average $[Na^+]_p$ of a given cell was not correlated with its resting membrane potential (n/N = 24; Pearson: − 0.11, $p$ = 0.59). To test if dialysis by the pipette saline influenced $[Na^+]_p$, $[Na^+]_p$ was first determined in cells held in whole-cell mode, before the patch-pipette was gently withdrawn. After cells were allowed to reseal for at least 15 min,

average $[Na^+]_p$ was not different (whole-cell: 17.0 ± 3.6 mM; pipette withdrawn: mean: 17.5 ± 11.7 mM; n/N = 7; $p$ = 0.93) (Fig. 3c). This result shows that $[Na^+]_p$ is apparently not clamped by the pipette saline ($[Na^+]$: 11.2 mM), similar to what was observed in neuronal dendrites[25].

As an alternative approach for the determination of $[Na^+]_p$, we revisited experiments performed in bolus-loaded slices. After drawing ROIs around processes of individual cells using SR101 images, ING-2 FL was evaluated from the same focal plane (Fig. 3d and Supplementary Fig. 4). Average $[Na^+]_p$ of single cells determined in bolus-loaded slices

**Fig. 1 | MP-FLIM-based determination of astrocyte [Na⁺] in the CA1 area.**
**a** Experimental design. **b** Z-stacks of SR101 fluorescence intensity and ING-2 FL. *Left*: 3D-reconstruction of SR101 (depth: 34-94 μm). *s.p.: stratum pyramidale, s.r.: stratum radiatum*, scale: 20x20x60 μm. *Right:* Two optical planes (68, 94 μm) of the same slice. 1ˢᵗ column: ING-2 FL; 2ⁿᵈ: SR101 intensity of the boxed region; 3ʳᵈ: ING-2 FL (SR101 mask employed, see Methods). Numbers indicate FLs determined from somatic ROIs. Scales: 20 μm. **c** Scheme of calibration strategy, indicating changes in FL with changes in [Na⁺]. **d** FL images (SR101 mask employed) of a bolus-stained astrocyte in calibration salines containing different [Na⁺]. Color-code: average, amplitude-weighted FL ($\tau_{AVG}$). Scale: 10 μm. **e** Relation between $\tau_{AVG}$ and [Na⁺] in neuronal and astrocyte somata ($n = 417$, $N = 15$). Red line: Michaelis-Menten fit. Note that FL saturates ~ 1.8 ns/100 mM Na⁺. The gray shaded area highlights the range depicted on the right. Right: Relation between $\tau_{AVG}$ and [Na⁺] in astrocyte somata (blue; $n = 13$, $N = 4$) and processes (green; $n = 14$, $N = 4$) between 0-30 mM [Na⁺]. Red

dotted line: best fit of the full calibration depicted left. **f** FL (left) and somatic [Na⁺] (right) of astrocytes ($n = 369$, $N = 32$). **g** Correlation between [Na⁺]$_s$ and depth of the optical section ($n = 186$, $N = 7$; Pearson: 0.17, $R^2 = 0.03$). **h** [Na⁺]$_s$. Dotted gray line: single-Gaussian, blue and green line: double-Gaussian distribution ($n = 369$, $N = 32$). Note that the bimodal distribution fits better than the unimodal ($\Delta Chi^2 = 814$, F-test: $p < 0.0001$). **i** FL images (SR101 mask employed) of two astrocytes with initially low (top) and initially high (bottom) baseline [Na⁺]$_s$ in control and Na⁺-free calibration saline (0 Na⁺/Cal). Yellow numbers indicate [Na⁺]$_s$. Scale: 10 μm. **j** Left: Astrocytic [Na⁺]$_s$ in control and 0 Na⁺/Cal. Lines connect data points from individual cells ($n = 13$, $N = 4$; $p = 7.36E\text{-}06$). Right: Astrocytic and neuronal [Na⁺]$_s$ in Na⁺ free ACSF (0 Na⁺/ACSF) and 0 Na⁺/Cal. Lines connect data points from individual cells ($n = 121$, $N = 3$; $p = 0.3$). **f, g, j**: diamonds: individual data points, boxes: 25/75, whiskers: SD, lines: median, squares: mean. Further details on statistics are provided in the results and statistical summary file. Source data are provided as a Source Data file.

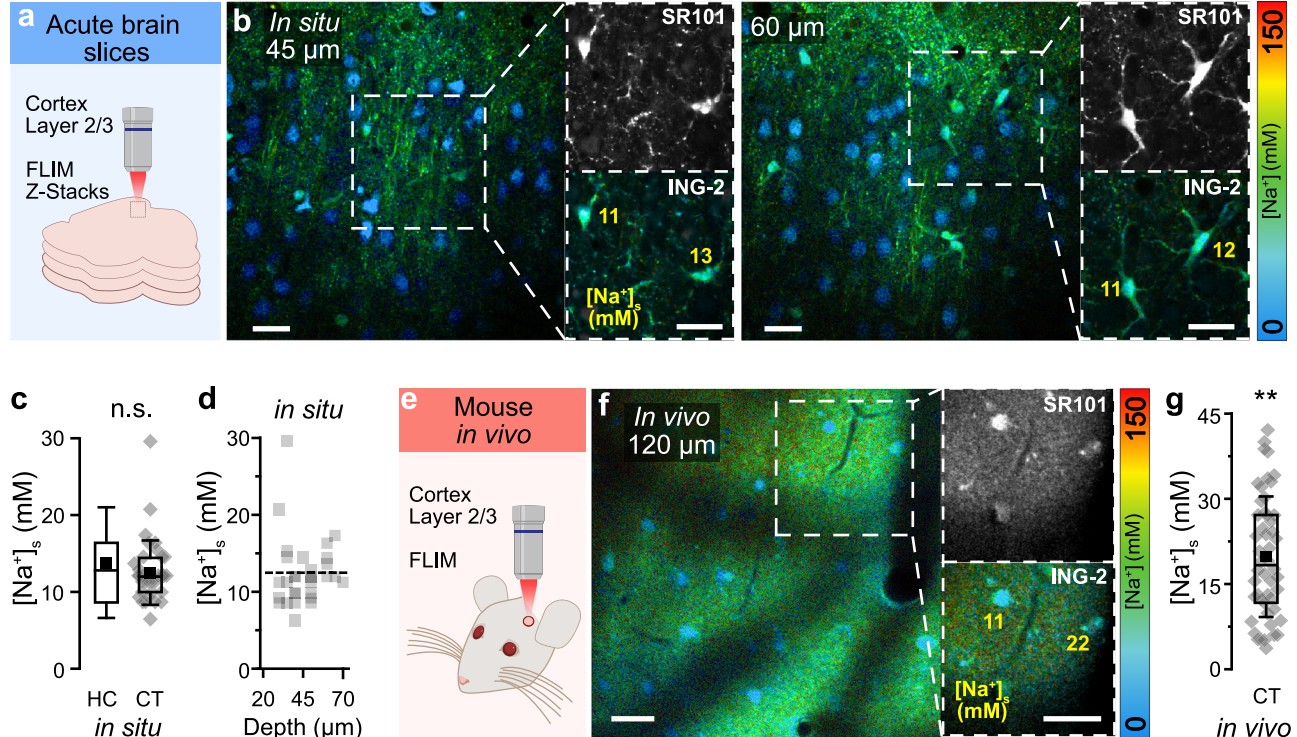

**Fig. 2 | [Na⁺] of cortical astrocytes in situ and in vivo. a** Scheme of experimental design of MP-FLIM in cortical tissue slices. **b** FL images of a cortical tissue slice at 45 μm (left) and 60 μm (right) of depth. Smaller images on the right depict the SR101 channel (top) and the ING-2 FL (masked with SR101) of the boxed regions at higher magnification. **c** [Na⁺]$_s$ in hippocampal (HC, data taken from Fig. 1f) and cortical slices (CT; $n = 34$, $N = 4$; $p = 0.53$). **d** Correlation between [Na⁺]$_s$ of cortical astrocytes and the depth of the optical section (Pearson: 0.17, $R^2 = 0.03$). **e** Schematic of MP-FLIM performed in vivo. **f** FL image at 120 μm of depth. Boxed

regions on the right show SR101 and color-coded ING-2 FL (masked with SR101) at higher magnification. **g** Estimated astrocytic [Na⁺]$_s$ in vivo (n = 50, 12 fields of view; vs. CT in situ: $p = 0.0017$). **b, f** Color-code for ING-2 FL is shown on the right. Yellow numbers indicate estimated [Na⁺]$_s$ in the indicated cells (in mM). **c, g** diamonds: individual data points, boxes: 25/75, whiskers: SD, lines: median, squares: mean. All scales: 20 μm. Further details on statistics are provided in the results and statistical summary file. Source data are provided as a Source Data file.

was $16.9 \pm 5.8$ (range: 6−30 mM; $n = 50$, $N = 6$). In the same set of astrocytes, [Na⁺]$_s$ was significantly lower ($12.4 \pm 5.1$ mM, range: 6−28 mM; $p = 5.34E\text{-}06$) (Fig. 3e). Notably, [Na⁺]$_p$ determined in bolus-loaded slices was similar to the average [Na⁺]$_p$ of cells dye-loaded by patch-clamp (compare Fig. 3b/3e; $p = 0.67$). This demonstrates that the reduced dye concentration in bolus-loaded processes did not distort the fitting of photon distributions and calculation of FL. Moreover, we found that in bolus-loaded astrocytes, [Na⁺]$_p$ in individual processes of a given cell was weakly linearly correlated with [Na⁺]$_s$ ($n_p = 122$, $n = 50$, $N = 6$; Pearson: 0.4, $p = 0.0038$, $R^2 = 0.16$) (Fig. 3f).

Next, we studied the variability of [Na⁺]$_p$ within a given astrocyte dye-loaded via a patch-pipette (Fig. 3g). Averaging 131 individual processes of 29 cells resulted in a mean [Na⁺]$_p$ of $16.9 \pm 7.2$ mM ($n_p = 131$, $n = 29$, $N = 25$) (Fig. 3h). Within the same astrocyte, [Na⁺]$_p$ varied on

average by $6.6 \pm 5.9$ mM between individual branches, differing by as much as 25 mM (Fig. 3i). In addition, [Na⁺]$_p$ was dependent on the distance from the soma. At a distance between 5-15 μm, average [Na⁺]$_p$ was $14.9 \pm 7.2$ mM ($n_p = 51$, $n = 20$, $N = 20$), while at 15−25 μm, average [Na⁺]$_p$ increased to $20.8 \pm 8.7$ mM ($n_p = 21$, $n = 14$, $N = 14$; $p = 0.005$) (Fig. 3j).

Altogether, these results demonstrate that the [Na⁺]$_p$ of a given astrocyte is significantly higher than that in the soma and increases with increasing distance from the soma. Notably, we also detected a large variability in [Na⁺]$_p$ between different processes of individual cells. Our experiments thus reveal a large intracellular heterogeneity in astrocyte [Na⁺], evident not only between somata and adjacent processes and between different processes of individual cells, but also within an individual process.

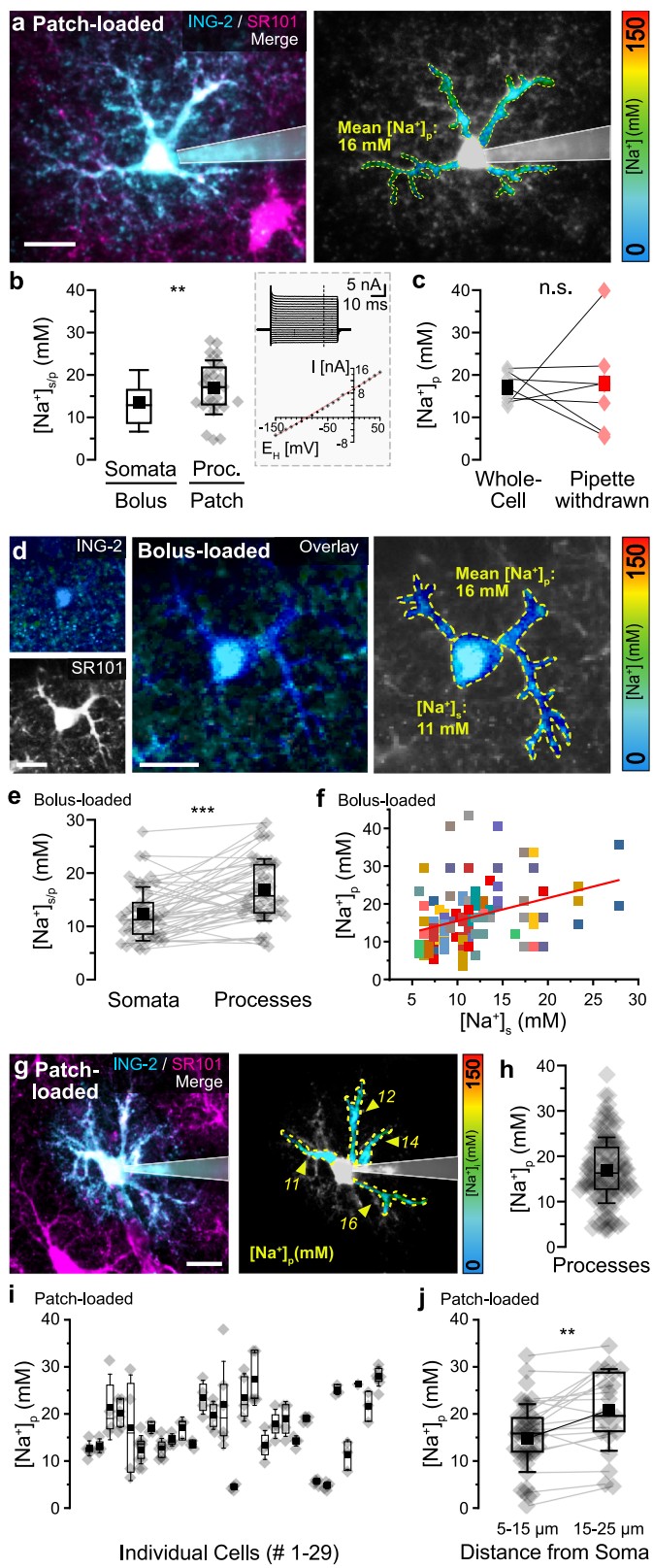

**Fig. 3 | [Na⁺] in processes of hippocampal astrocytes in situ. a** Left: Merge of ING-2 (cyan) and SR101 (magenta) intensity images of an astrocyte loaded with ING-2 via whole-cell patch-clamp. Right: color-coded ING-2 FL image. **b** [Na⁺]$_s$ in bolus-loaded slices (data from Fig. 1f) and average [Na⁺]$_p$ per cell determined from patch-clamped astrocytes (n/N = 29; p = 0.004). Inset: Typical I/V relationship of an SR101-positive cell held at − 85 mV and subjected to 10 ms voltage steps ranging from − 150 to + 50 mV. The I/V plot depicts the current amplitudes determined at the indicated dashed line, colored line: linear regression curve. **c** Average [Na⁺]$_p$ per cell determined in whole-cell mode and after withdrawal of the patch pipette (n/N = 7; p = 0.93). Lines connect data points from individual cells. **d** Left: Images of ING-FL and SR101 fluorescence of a bolus-loaded astrocyte. Center: ING-2 FL image (masked with SR101). Right: Color-coded image illustrating [Na⁺]$_p$ and [Na⁺]$_s$. **e** Paired average [Na⁺]$_p$ and [Na⁺]$_s$ of bolus-loaded astrocytes (n = 50, N = 6; p = 5.34E-06). **f** Relation between [Na⁺]$_s$ and [Na⁺]$_p$ (n$_p$ = 122, n = 50, N = 6; Pearson: 0.4, $R^2$ = 0.11). Symbols with the same color and same somatic [Na⁺] (i.e., arranged in a column) represent processes from the same astrocyte. **g** Left: Merge of ING-2 (cyan) and SR101 (magenta) intensity images of an astrocyte loaded via patch-clamp. Right: color-coded ING-2 FL image. **h** [Na⁺]$_p$ as determined in individual processes (n$_p$ = 131, n = 29, N = 25). **i** [Na⁺]$_p$ depicted for individuals cells (#1-29). **j** [Na⁺]$_p$ derived from ROIs at 5-15 µm (n$_p$ = 51, n/N = 20) and 15−25 µm (n$_p$ = 21, n/N = 14) from the soma. Lines connect data points from the same process (p = 0.00451). **a**, **d**, **g** Color-code for ING-2 FL is shown on the right. Dotted lines indicate ROIs for FL determination, yellow numbers show [Na⁺] in the respective ROIs. **b**, **c**, **e**, **h**, **i**, **j** diamonds: individual data points, boxes: 25/75, whiskers: SD, lines: median, squares: mean. All scales: 20 µm. Further details on statistics are provided in the results and statistical summary file. Source data are provided as a Source Data file.

The same was true for processes ([Na⁺]$_p$: control: mean 19.2 ± 8.9 mM, range 6−46 mM, n$_p$ = 44, n = 13, N = 4; TTX: mean 16.1 ± 6.4 mM, range 4−30 mM; n$_p$ = 49, n = 13, N = 4; p = 0.174) (Fig. 4a). Furthermore, the bimodal distribution of [Na⁺]$_s$ persisted in TTX (ΔChi² = 20, F-test: p < 0.0001) (Fig. 4b). These results indicate that spontaneous activity of microcircuits resulting in the local release of K⁺ and glutamate is not primarily responsible for the different Na⁺ levels nor its bimodal distribution in astrocytes. Of note, release of glutamate will activate Na⁺ import by EAATs, while release of K⁺ will result in activation of NKA and Na⁺ export, counteracting any EAAT-related Na⁺ influx.

The potential involvement of gap junctions was studied by bath application of the gap junction blocker carbenoxolone (CBX; 100 µM). Upon perfusion of CBX, the mean [Na⁺]$_s$ of astrocytes as well as its range increased significantly (control: mean 13.0 ± 7.4 mM, range 2.3−45.4 mM; n = 186, N = 7; CBX: mean 20.0 ± 13.3 mM, range 3.5−96.6 mM; n = 134, N = 7; p = 0.03) (Fig. 4c, d). CBX did not alter the FL of SR101, indicating that it did not cause Na⁺-independent changes in the FL of ING-2 (Supplementary Fig. 5). Similar to mice deficient for astroglial Cx43 and Cx30[31], we also found that CBX increased [K⁺]$_e$ slightly from 2.48 ± 0.1 mM to 2.54 ± 0.12 mM as measured using ion-selective microelectrodes (N = 8; p = 0.016) (Fig. 4e and Supplementary Fig. 6). At the same time, the bimodal distribution of [Na⁺]$_s$ was maintained (ΔChi² = 67, F-test: p < 0.001) (Fig. 4f). Furthermore, we analyzed the difference in [Na⁺]$_s$ between neighboring astrocytes in relation to their distance in a given slice in control conditions (i.e., without CBX) (644 connections between 67 cells, N = 4) (Fig. 4g–i). We found that the variance of [Na⁺]$_s$ increased significantly with increasing distance (0-50 µm: 17 (92 connections), 50−100 µm: 54 (272 connections), p < 0.0001; >100 µm: 72 (280 connections), p = 0.01823) (Fig. 4i). This indicates that the [Na⁺]$_s$ of an astrocyte is at least partly dependent on its spatial position or whether it belongs to a special gap junction-coupled syncytium.

We next studied the effect of an inhibition of the NKA by perfusing slices with nominally K⁺-free saline for 2 minutes ("low K⁺") (Fig. 4j–n). This caused [K⁺]$_e$ to transiently decrease to ~1.5 mM (N = 8) (Supplementary Fig. 6). In pyramidal neurons, low K⁺ caused a hyperpolarization from an E$_m$ of − 68.2 ± 7.2 mV to − 76.4 ± 5.7 mV (n = 9, N = 7) (Fig. 4k). Upon restoring standard [K⁺]$_e$, neurons rapidly repolarized

## Determinants of baseline [Na⁺] of astrocytes

To analyze determinants of astrocytic baseline [Na⁺], we performed dynamic MP-FLIM in bolus-loaded hippocampal slices. Bath application of tetrodotoxin (TTX; 0.5 µM) to block spontaneous action potential activity did neither affect mean somatic [Na⁺]$_s$ nor its range ([Na⁺]$_s$: control 12.5 ± 3.6 mM, range 6−23 mM; n = 60, N = 18; TTX: mean 11.9 ± 4.1 mM, range 5−24 mM; n = 53, N = 22; p = 0.44) (Fig. 4a).

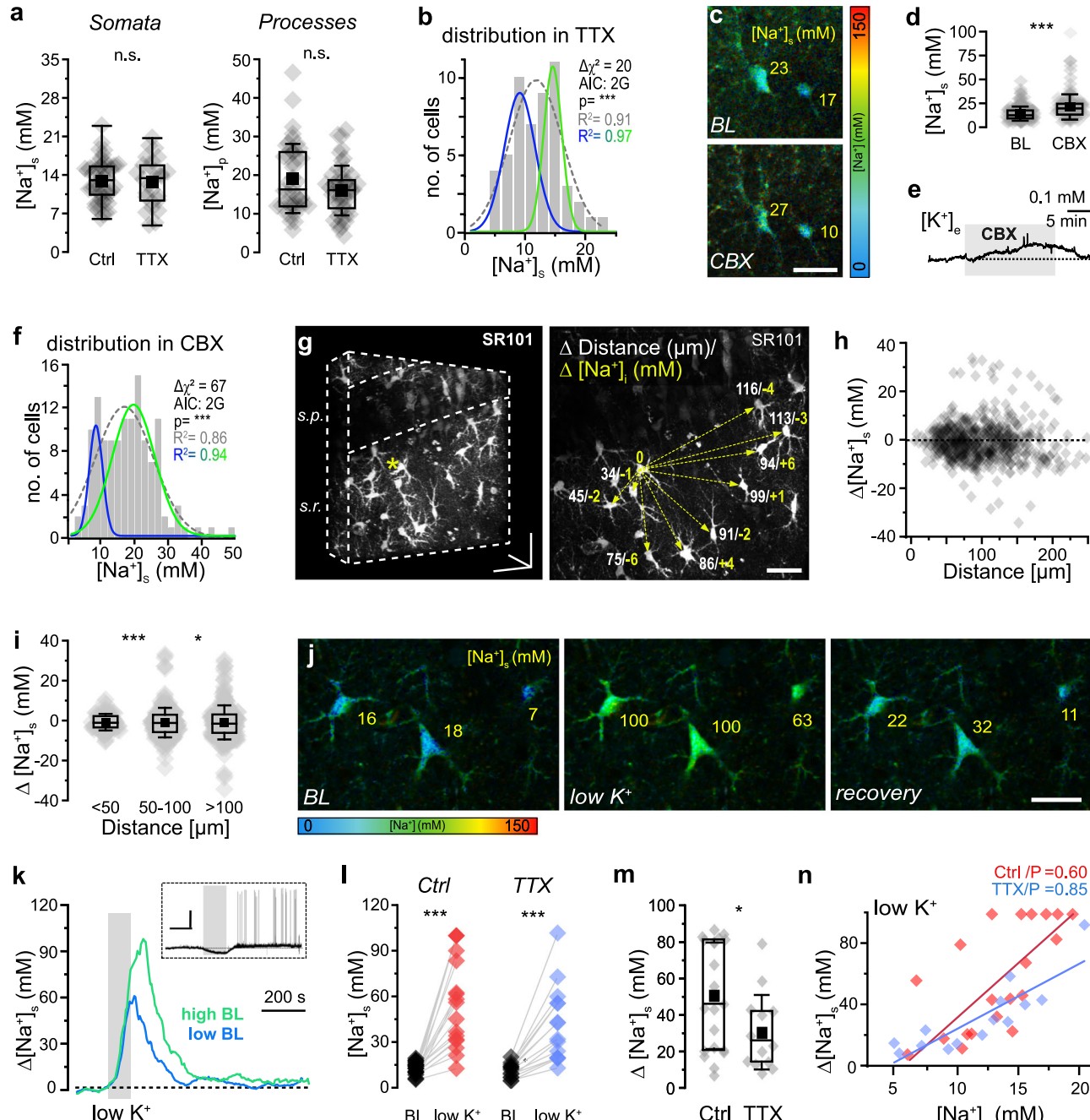

**Fig. 4 | Determinants of astrocytic [Na⁺].** **a** [Na⁺]$_s$ in control (Ctrl; $n = 60$, $N = 18$) and TTX ($n = 53$, $N = 22$; $p = 0.44$). Right: same for processes (Ctrl: $n_p = 44$, $n = 13$, $N = 4$; TTX: $n_p = 49$, $n = 13$, $N = 4$; $p = 0.174$). **b** [Na⁺]$_s$ in TTX. Dotted gray line: single-Gaussian, blue and green line: double-Gaussian distribution (ΔChi² = 20, F-test: $p < 0.0001$). **c** FL images (SR101-masked) depicting astrocytes at baseline (BL) and with carbenoxolone (CBX). Yellow numbers: [Na⁺]$_s$, color-code on the right. **d** [Na⁺]$_s$ at baseline ($n = 186$, $N = 7$) and with carbenoxolone ($n = 134$, $N = 7$; $p = 1.87$E-09). **e** Carbenoxolone-induced change in [K⁺]$_e$ (gray area; avg. 8 measurements in 8 slices). **f** [Na⁺]$_s$ in CBX ($n = 134$, $N = 7$). Dotted gray line: single-Gaussian, blue and green line: double-Gaussian distribution (ΔChi² = 67, F-test: $p < 0.001$). **g** 3D-reconstruction of SR101 fluorescence (left) and corresponding maximum intensity projection (right). Distance from astrocyte marked "*/0" indicated in white, Δ[Na⁺]$_s$ in yellow. **h, i** Correlation between Δ[Na⁺]$_s$ of astrocytes in relation to their distance in 3D. **h** 644 connections, **i** 0-50 μm: 92 connections, 50–100 μm: 272, >100 μm:

280; each $N = 4$; **i**: $p < 0.0001$ (***), $p = 0.01823$ (*). **j–n** Effect of K⁺-free saline ("low K⁺") for 2 minutes. **j** FL images (SR101-masked) showing baseline [Na⁺]$_s$, peak [Na⁺]$_s$ in low K⁺ and [Na⁺]$_s$ after recovery. Yellow numbers: [Na⁺]$_s$, color-code at bottom. **k** Smoothed (rolling avg, 5 pts) [Na⁺]$_s$ changes in two astrocytes with low (7.1 mM, blue) and high (16.3 mM, green) baseline [Na⁺]$_s$. Inset: Changes in the neuronal E$_m$ induced by low K⁺ (gray shaded area). Scales: 100 s/20 mV. **l** Left: baseline [Na⁺]$_s$ and peak [Na⁺]$_s$ in low K⁺. Lines connect data points from individual cells ($n = 24$, $N = 4$; $p = 1.19$E-07). Right: same in TTX ($n = 14$, $N = 5$; $p = 8.87$E-05). **m** Peak changes in [Na⁺]$_s$ in low K⁺ in control and TTX ($p = 0.014$). **n** Correlation between baseline [Na⁺]$_s$ and peak [Na⁺]$_s$ changes in low K⁺ in control (red) and TTX (blue) (Ctrl: Pearson: 0.60, TTX: Pearson: 0.85). **a, d, h, i, l, m, n** diamonds: individual data points, boxes: 25/75, whiskers: SD, lines: median, squares: mean. All scales: 20 μm. Further details on statistics are provided in the results and statistical summary file. Source data are provided as a Source Data file.

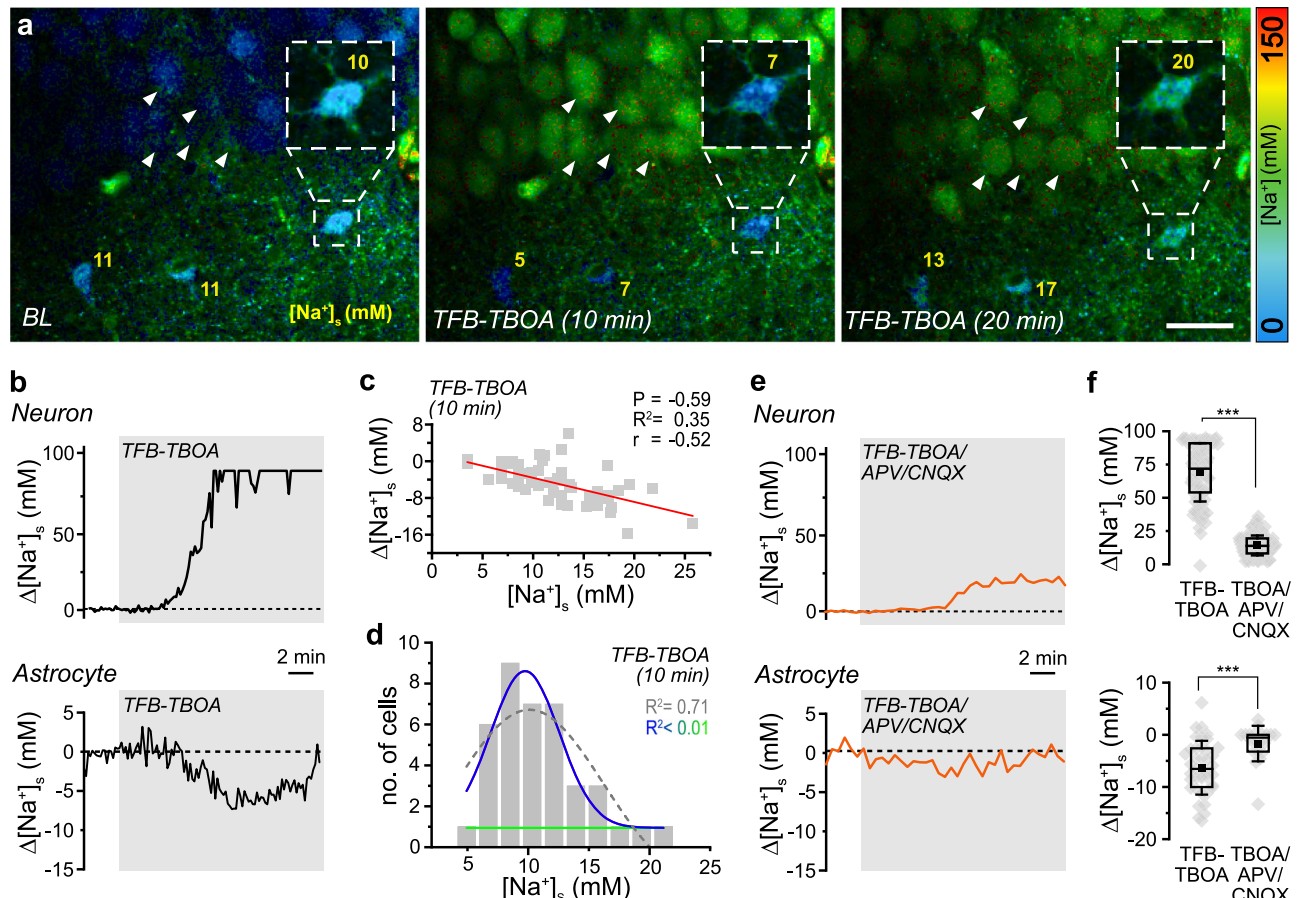

**Fig. 5 | Role of Na⁺-dependent glutamate transporters. a** FL images of an experiment showing baseline [Na⁺]$_s$ (BL), and changes in [Na⁺]$_s$ after perfusion with TFB-TBOA to inhibit EAATs for 10 and 20 minutes. The box shows the ING-2 FL (masked with SR101) of an astrocyte at 1.5 x magnification. Note that the cell bodies of CA1 neurons visible in the upper part load with Na⁺ and round up (arrowheads) in TFB-TBOA. Color-code for ING-2 FL is shown on the right, yellow numbers indicate [Na⁺] in the indicated somata. Scale 20 μm. **b** Traces depicting changes in [Na⁺]$_s$ induced by TFB-TBOA (gray shaded area) in a CA1 neuron (top) and an astrocyte (bottom). **c** Correlation between initial baseline astrocytic [Na⁺]$_s$ and the peak decrease in [Na⁺]$_s$ induced by TFB-TBOA ($n = 60$, $N = 18$). Red line: linear fit of the data (slope: − 0.52, Pearson: − 0.59, $R^2 = 0.35$). Squares represent individual data points. **d** [Na⁺]$_s$ after ≥ 20 min of perfusion with TFB-TBOA ($n = 60$, $N = 18$). Dotted

gray line: single-Gaussian, blue and green line: double-Gaussian distribution. Note that fitting a double-Gaussian was not possible ($R^2 < 0.01$). **e** Traces depicting changes in [Na⁺]$_s$ induced by TFB-TBOA (gray shaded area) in a CA1 neuron (top) and an astrocyte (bottom) in the presence of the NMDA and AMPA receptor antagonists CNQX and APV for ≥ 15 min. **f** TFB-TBOA-induced peak changes in neuronal [Na⁺]$_s$ (top) (TBOA: $n = 96$, $N = 4$; TBOA/CNQX/APV: $n = 131$, $N = 7$; $p < 1E-10$) and astrocyte [Na⁺]$_s$ (bottom) (TBOA: $n = 60$, $N = 18$; TBOA/CNQX/APV: $n = 20$, $N = 12$; $p = 0.000683$) with and without CNQX and APV. Diamonds: individual data points, boxes: 25/75, whiskers: SD, lines: median, squares: mean. Further details on statistics are provided in the results and statistical summary file. Source data are provided as a Source Data file.

and then experienced a phase of increased activity (Fig. 4k). Low K⁺ caused astrocytic [Na⁺]$_s$ to increase to 69.1 ± 31.9 mM (range: 13–133 mM; $n = 24$, $N = 4$; $p = 1.19E-07$) (Fig. 4j–n). At the same time, the FL of SR101 was not altered, indicating that low K⁺ did not cause unspecific, Na⁺-independent changes in FL (Supplementary Fig. 5). The peak [Na⁺]$_s$ increase occurred up to 2 minutes after standard [K⁺]$_e$ was restored (Fig. 4k). Its magnitude was linearly correlated with the initial baseline [Na⁺]$_s$ (Pearson: 0.60, $p = 0.0023$), with astrocytes having a high baseline [Na⁺]$_s$ showing a stronger increase than those with a low baseline (Fig. 4k–n). Recovery from low K⁺-induced Na⁺ load followed a monoexponential decay (mean decay time constant: 77 s for cells with an initial baseline [Na⁺]$_s$ between 6–10 mM ($n = 5$), 86 s for 10.1–15 mM ($n = 9$), and 97 s for 15.1–20 mM ($n = 5$)).

Application of TTX (0.5 μM) did not alter neuronal $E_m$ (mean: − 72.6 ± 4.4 mV; $p = 0.202$) nor the $E_m$ during low K⁺ as compared to control (mean: − 77.4 ± 3.5 mV; $p = 0.709$), but prevented the phase of increased neuronal activity and action potential firing upon restoring standard [K⁺]$_e$ ($n = 6$, $N = 5$). In TTX, [Na⁺]$_s$ increased to 42.5 ± 24.7 mM during low K⁺ (range: 13–100 mM; $n = 14$, $N = 5$; $p = 8.87E-05$) (Fig. 4l), which was a smaller change than in control ($p = 0.014$) (Fig. 4m). The

positive linear correlation between initial baseline [Na⁺]$_s$ and [Na⁺] increase was maintained in TTX (Pearson: 0.85, $p < 0.0001$) (Fig. 4n). This data shows that transient inhibition of the NKA by low [K⁺]$_e$ results in strong Na⁺ influx into astrocytes, which is promoted by increased neural activity after normal [K⁺]$_e$ is restored.

Uptake of glutamate by EAATs is a major contributor to activity-induced [Na⁺] increases in astrocytes[7]. To test if EAAT activity contributes to setting astrocytic baseline [Na⁺], slices were perfused with the EAAT-inhibitor TFB-TBOA (1 μM) (Fig. 5a–d). This caused a transient [K⁺]$_e$ elevation by 2.3 mM ($N = 10$; $p = 0.000183$) (Supplementary Fig. 6). Moreover, TFB-TBOA, while not affecting the FL of SR101 (Supplementary Fig. 5), resulted in a strong, sustained change in [Na⁺]$_s$ in somata of pyramidal neurons, which reached the saturation limit of ING-2 (~100 mM) within 6–10 min ($n = 91$, $N = 4$) (Fig. 5a, b). It was accompanied by a rounding up and swelling of neuronal somata (Fig. 5a and Supplementary Fig. 7). In contrast, TFB-TBOA caused a slow initial decrease in astrocytic [Na⁺]$_s$ by on average 4.6 ± 4.0 mM ($n = 60$, $N = 18$) with no detectable somatic swelling (Fig. 5a, b). Changes in astrocytic [Na⁺]$_s$ were weakly correlated with the initial baseline, and cells with a higher baseline exhibited a greater decrease than those

with a lower baseline (slope: -0.52, Pearson: $-0.59$, $p < 0.0001$) (Fig. 5c). As opposed to TTX or CBX, astrocytic $[Na^+]_s$ shifted towards a unimodal distribution in TFB-TBOA ($R^2 = 0.71$) (Fig. 5d), indicating that EAAT-related $Na^+$ influx contributes to the biphasic distribution of $[Na^+]_s$.

Inhibition of EAATs causes an immediate increase in extracellular glutamate. To address the relevance of ionotropic glutamate receptors in $[Na^+]_s$ elevations, we exposed slices to the receptor blockers CNQX (10 μM) and APV (50 μM). Under these conditions, the TFB-TBOA-induced changes in neuronal and astrocytic $[Na^+]_s$ were significantly reduced, albeit not entirely blocked (neurons: $n = 131$, $N = 4$; astrocytes: $n = 20$, $N = 7$) (Fig. 5e, f), as was the swelling of neuronal somata ($n = 101$, $N = 4$) (Supplementary Fig. 7). These experiments emphasize the prominent role of ionotropic glutamate receptors in neuronal loading $Na^+$ and swelling in the presence of EAAT blockers[26]. They also demonstrate that, during the initial stages of neuronal $Na^+$ loading, astrocytes can maintain low $[Na^+]_s$.

Taken together, these results show that spontaneous action-potential activity has no detectable influence on baseline $[Na^+]_s$ of hippocampal astrocytes in an acute slice preparation and confirm the expected vital influence of the NKA on astrocytic $[Na^+]_s$. Moreover, our results indicate that the baseline $[Na^+]_s$ of astrocytes is at least partially determined by $Na^+$ influx through EAATs. The data also suggests that cells with a high baseline $[Na^+]_s$ are subject to a strong relative influx of $Na^+$, whereas cells with a low baseline $[Na^+]_s$ have relatively weaker $Na^+$ influx. Alternatively, and/or in addition, high-baseline $[Na^+]_s$ cells might express lower levels of the NKA, or display lower NKA activity in response to a $Na^+$ load.

## $[Na^+]$ determines uptake of extracellular $K^+$ by astrocytes

Uptake of $K^+$ by the astrocytic NKA plays a vital role for $[K^+]_e$ homeostasis[4]. To study the influence of increased $[K^+]_e$ on astrocytic $[Na^+]$, slices were perfused with ACSF containing 10 mM $K^+$ ("high $K^+$") for 2 min. This resulted in a transient $[K^+]_e$ increase to $7.3 \pm 0.9$ mM ($N = 8$), which was significantly higher than that observed with TFB-TBOA ($p = 0.00017$) (Supplementary Fig. 6). In bolus-loaded astrocytes, high $K^+$ caused an average decline of $[Na^+]_s$ to $6.3 \pm 1.6$ mM ($n = 29$, $N = 8$) (Fig. 6a–f). In individual cells, the magnitude of this decrease showed a strong linear correlation with their former baseline (slope: $-0.63$; Pearson: $-0.93$, $p < 0.0001$) (Fig. 6g). The amplitude of high $K^+$-induced changes in $[K^+]_e$ was not influenced by TTX (mean: $7.1 \pm 0.49$ mM; $N = 5$; $p = 0.77$) (Supplementary Fig. 6). Moreover, TTX did not alter high $K^+$-induced changes in astrocytic $[Na^+]_s$ (mean: $6.6 \pm 2.1$; $n = 13$, $N = 4$; $p = 0.2$), nor the linear correlation between initial $[Na^+]_s$ and peak decline (slope: $-0.54$; Pearson: $-0.87$, $p < 0.0001$) (Fig. 6d–f). This indicates that the $K^+$-induced activation of the NKA in astrocyte somata dominated over any $Na^+$ influx, potentially resulting from increased neuronal activity and/or activation of $Na^+$ inward transporters such as EAATs.

To analyze the influence of high $K^+$ on $[Na^+]$ in individual processes, astrocytes were dye-filled by whole-cell patch-clamp (Fig. 6h, i). Increasing $[K^+]_e$ caused an average decrease of $[Na^+]_p$ from $23.3 \pm 6.5$ mM to $9.5 \pm 3.7$ mM (Fig. 6j, k), which was significantly larger compared to the soma ($n_p = 17$, $n = 5$, $N = 5$; $p < 0.0001$). During perfusion of TTX, high $K^+$ reduced $[Na^+]_p$ from $16.6 \pm 6.5$ mM to $7.0 \pm 3.4$ mM ($n_p = 27$, $n = 5$, $N = 5$; $p = 1.28E-12$) (Fig. 6l). TTX thus decreased the mean amplitude of the high $K^+$-induced decline in $[Na^+]_p$ ($p = 0.009$) (Fig. 6m), indicating that high $K^+$ induced action-potential-related neuronal activity promoted the drop in astrocyte $[Na^+]_p$. Average $[Na^+]_p$ and high $K^+$-induced decline in $[Na^+]_p$ followed a linear correlation with comparable slopes in the presence and absence of TTX (control: slope: $-0.68$, Pearson: $-0.81$, $p < 0.0001$; TTX: slope: $-0.55$, Pearson: $-0.87$, $p = 0.00013$) (Fig. 6n).

In summary, these results show that an increase in $[K^+]_e$ results in an immediate decrease in $[Na^+]$ in somata and processes of astrocytes,

the magnitude of which is strongly correlated with the initial baseline $[Na^+]$. As $Na^+$ export is virtually solely mediated by the NKA, this suggests that compartments with a high baseline $[Na^+]$ undergo a stronger $K^+$-induced activation of the NKA than those with a low $[Na^+]$.

## Expression of NKA subunits

Astrocytes express different NKA subunits, which differ in their binding affinities for internal $Na^+$ and external $K^+$[4]. Recent RNA sequencing showed predominant expression of α2β2 in about 70% of mouse forebrain astrocytes, while about 30% expressed α2β1[27,32,33]. To gain insight in the expression levels and spatial expression patterns of β1 and β2 subunits in the CA1 region, we probed hippocampal sections for mRNA using RNAscope[34] and for protein expression using immunohistochemistry (IHC) (Supplementary Tables 1–4).

In the pyramidal cell (PC) layer, RNAscope for ATP1b1 (β1) showed high neuronal and/or perineuronal labeling intensities, while RNA levels for ATP1b2 (β2) were low ($N = 3$) (Fig. 7a). In the *stratum radiatum*, there was lower ATP1b1 labeling, but this clearly comprised GFAP-positive cells. In contrast, GFAP-positive cells in the *stratum radiatum* were intensively labeled for ATP1b2 (Fig. 7a). Generally, β2 signals were high in both somata and processes of GFAP-positive cells, while β1 signals were mainly detectable in somata and some proximal processes. For both probes, we also identified cells solely labeled either for GFAP or by RNAscope. Comparing RNA-labeling for β1 and β2 (Fig. 7a, lower panel) moreover showed that most GFAP-positive cells were double-positive, but occasionally, cells were found which were positive for only one of them.

When comparing the IHC signals of β1 and β2, neuronal and/or perineuronal labeling in the PC layer was slightly higher in β2 than in β1-marked cells (Fig. 7b and Supplementary Fig. 8a–d). In the *stratum radiatum*, similar patterns were found as for RNA signals. The majority of GFAP-positive cells were positive for both subunits in IHC, but also single GFAP-labeled cells, positive for only one of the subunits, were found (Supplementary Fig. 8a,c). IHC-labels for both β1 and β2 were detected in GFAP-expressing somata, and at least in proximal processes (Fig. 7b). Clear, above-threshold anti-β2 labeling was apparent in a greater proportion of GFAP-positive cells than anti-β1 labeling. Accordingly, the overall proportion of β2-labeled processes also seemed to be higher as compared to β1.

Overall, these results demonstrate the presence of NKA β1 and β2 subunits at the mRNA and protein levels in GFAP-positive cells, confirming their expression in astrocytes. Generally, the mRNA and protein staining patterns align with a lower level of β1 compared to β2. Furthermore, the staining patterns suggest that processes overall exhibit higher expression of β2 than β1.

## Biophysical modeling of astrocytic $[Na^+]$

To study the influence of specific subunit compositions on baseline $[Na^+]$ and changes in $[Na^+]$, we performed mathematical simulations. For a fixed NKA expression and $Na^+$ influx, these simulations predict a $[Na^+]$ of 11.8 mM for astrocytes with α2β1 subunit only (Fig. 8a). Exclusive expression of α2β2 resulted in a $[Na^+]$ of 22.3 mM, whereas mixing both isoforms resulted in levels in between (Fig. 8a). Next, we fixed the isoforms to a distribution of 30% α2β1 and 70% α2β2[32,33], and varied the overall NKA expression level, i.e., pump strength, between 60–180%. $[Na^+]$ increased with decreasing NKA expression from -8 mM at 180% to > 40 mM at 60% (Fig. 8a). Varying the strength of $Na^+$ influx between 50–170% at fixed NKA composition and expression, caused $[Na^+]$ to increase to near 40 mM (Fig. 8a). When changing the subunit composition and/or expression of NKA together with the strength of $Na^+$ influx, the effect on $[Na^+]$ was strongest in cells predominantly expressing α2β2 (Supplementary Table 5). For example, decreasing $Na^+$ influx to 60% decreased $[Na^+]$ of astrocytes with 100% α2β2 expression to levels close to those of astrocytes that exclusively express α2β1. This led to a shift to the left and a narrowing of the $[Na^+]$

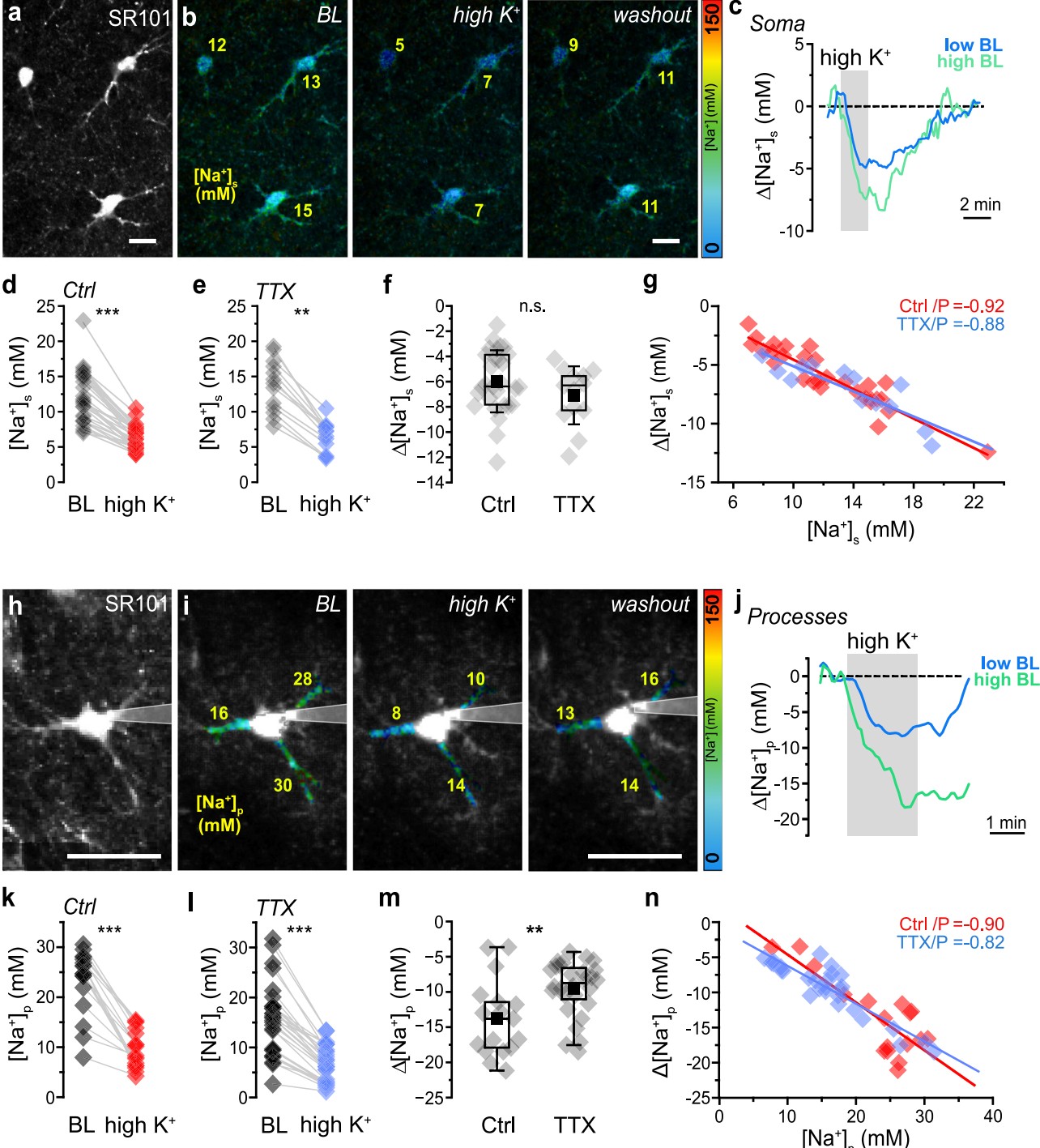

**Fig. 6 | Changes in astrocytic [Na⁺] induced by increasing external K⁺. a-n** Effect of perfusion with saline containing 10 mM K⁺ for 2 minutes ("high K⁺") on [Na⁺] in somata ([Na⁺]$_s$, a-g) and processes ([Na⁺]$_p$, h-n). **a** SR101 image depicting several astrocytes. **b** FL images (masked with SR101) showing baseline [Na⁺]$_s$ (BL), peak decrease in [Na⁺]$_s$ induced by high K⁺, and [Na⁺]$_s$ after washout of high K⁺ for 5 minutes in three astrocytes. Color-code is shown on the right, yellow numbers indicate [Na⁺] in the indicated somata. Scale: 10 μm. **c** Traces showing changes in [Na⁺]$_s$ induced by high K⁺ (gray area) in two astrocytes with initially low (8.9 mM, blue trace) and initially high (15.1 mM, green trace) [Na⁺]$_s$. **d** [Na⁺]$_s$ in astrocytes at baseline (BL) and peak [Na⁺]$_s$ induced by high K⁺ ($p = 2.01E-13$). Lines connect data points from individual cells. **e** same as in d but with TTX present ($p = 1.16E-07$). **f** Peak decrease [Na⁺]$_s$ induced by high K⁺ in the absence (Ctrl) and presence of TTX ($p = 0.21$). **g** Correlation between initial baseline [Na⁺]$_s$ and the peak decrease in

[Na⁺]$_s$ induced by high K⁺ without (Ctrl, red) and with TTX (blue). Lines represent linear fits of the data (Ctrl: Pearson:-0.92, slope -0.63; TTX: Pearson:-0.88, slope:-0.54). **a-g:** Ctrl: $n = 29$, $N = 8$; TTX: $n = 13$, $N = 4$. **h-n** Same as in a-g, but for processes ([Na⁺]$_p$) from astrocytes dye-loaded by whole-cell patch-clamp. **h, i** Scale: 20 μm. **i** Time series (binning 5) of summed extended focus FL-images (3 z-steps, 3.5 μm step size). The right image shows [Na⁺]$_s$ after washout of high K⁺ for 2 min. **h, i** Scale: 20 μm. **j** Traces from two processes with initially low (16 mM, blue) and high (28 mM, green) baseline [Na⁺]$_p$. **k, l** Ctrl: $p = 0.000321$; TTX: $p = 1.28E-12$. **m** $p = 0.00861$. **n** Ctrl: Pearson:−0.90, slope:-0.68; TTX: Pearson:-0.82, slope:-0.55. **h–n:** Ctrl: $n_p = 17$, $n = 5$, $N = 5$; TTX: $n_p = 27$, $n = 5$, $N = 5$. **d–g, k–n:** diamonds: individual data points, boxes: 25/75, whiskers: SD, lines: median, squares: mean. Details on statistics are provided in the results and statistical summary file. Source data are provided as a Source Data file.

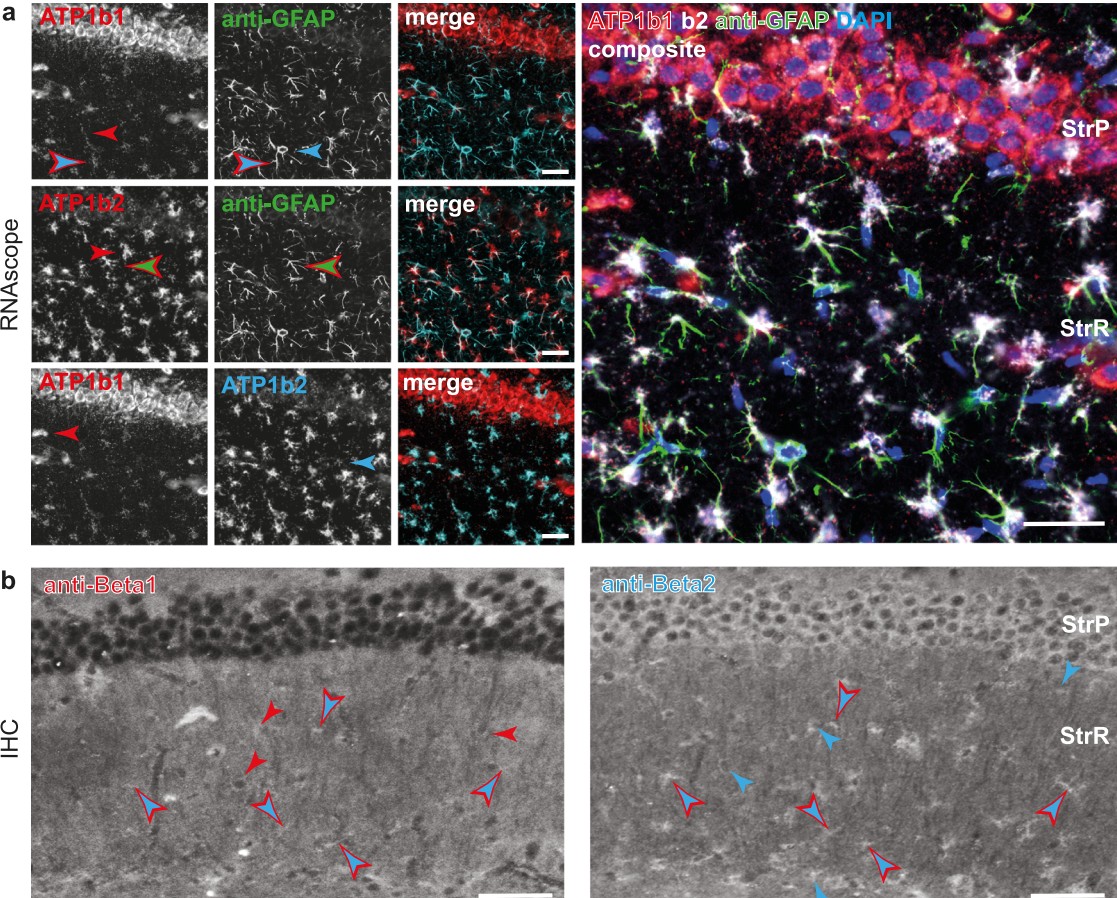

**Fig. 7 | Spatial expression patterns of NKA β1 and β2 subunits in the mouse CA1 region. a** RNAscope for ATP1b1 and ATP1b2. Left, upper and middle panel: Double-labeling of mRNA for subunits NKA β1 or β2 with immunohistochemical labeling for GFAP and merge of the two channels. Left, lower panel: Same tissue section, showing double-labeling of mRNA for β1 and β2 and merge. Right: Composite image of tissue section on the left, showing labeling for ATP1b1, ATP1b2, GFAP and DAPI. Please note that in all stainings, only a small proportion of cells are positive for one of the NKA subunit markers, while the majority of labeled cells are positive for both. Similar results were independently obtained at least 3 times for each experimental condition. **b** Subsequential immunohistochemical double-labeling for the NKA subunits β1 and β2. Note the different levels of perineuronal/neuronal labeling of both subunits in the *stratum pyramidale*. Similar results were independently obtained at least 3 times for each experimental condition. **a, b** single-colored arrowheads point out cells exhibiting only one label, double-colored arrowheads show cells labeled by both labels used. StrP: *stratum pyramidale*, StrR: *stratum radiatum*. **a, b** All scale bars: 80 μm.

distribution, comparable to that observed experimentally when EAATs were inhibited by TFB-TBOA (see Fig. 5d). Finally, we found that a combination of different NKA isoform composition and pump strength can replicate the bimodal distribution of astrocyte $[Na^+]_s$ reported experimentally (Fig. 8a). In addition, adding variable $Na^+$ influx to this would also result in a biphasic distribution.

Next, we tested how a specific NKA composition influences the astrocytes' response to NKA inhibition by subjecting them to 1 mM $[K^+]_e$ for 2 min. Astrocytes with exclusive α2β1 expression showed a $[Na^+]$ increase by ~55 mM, while cells with α2β2 experienced an increase by >100 mM (Fig. 8b). At rest, the flux through α2β1 is higher than through α2β2, resulting in lower $[Na^+]$ (Fig. 8b). In low $K^+$, the flux through α2β1 first increases as $[Na^+]$ rises, followed by a smaller decrease due to the drop in $[K^+]_e$ as compared to α2β2 (Fig. 8b).

When analyzing the response to increases in $[K^+]_e$, an opposite behavior was seen (Fig. 8c). The flux through α2β2 increases as $[K^+]_e$ increases, causing $[Na^+]$ to drop more. α2β1, on the other hand, is already close to saturation, and its flux increases only slightly as $[K^+]_e$ rises. However, it decreases sharply upon the drop in $[Na^+]$, countering a further decrease. Overall, this leads to a smaller decrease in $[Na^+]$ in α2β1-expressing astrocytes (Fig. 8c). Supplementary Fig. 9 and the interactive 3D-Plots (Link) illustrate how the flux through the two isoforms changes as we vary $[K^+]_e$ and $[Na^+]_i$ further. They also illustrate

NKA's dependency on $[Na^+]_i$, showing that a decline in $[Na^+]_i$ to 2 mM decreases its activity to below 2% in α2β1 and 10% in α2β2 expressing astrocytes, respectively.

The experimentally observed subcellular heterogeneity in astrocytic $[Na^+]$ could also be replicated by making the NKA expression and/or the $Na^+$ influx through EAAT as functions of distance from the soma (Fig. 8d). We modeled a reconstructed astrocyte[35] (RRID:SCR_002145; morphology #NMO_282188) with fixed NKA expression and isoform composition, but varied $Na^+$ influx by linearly increasing EAAT density in the processes from 100% (close to soma) to 300% (terminal region). Second, we fixed $[Na^+]$ influx and NKA strength but varied the ratio of α2β1/α2β2 along processes (α2β1-dominant near soma, α2β2-dominant near terminal). The first simulation shows a $[Na^+]_p$ of 12 mM near the soma and of 32.6 mM in most distal branches, while the second predicts a $[Na^+]_p$ of 11.9 mM near the soma and of 22.3 mM near the end of longest branch (Fig. 8d). When simulating global high $[K^+]_e$ for the latter scenario, $[Na^+]_p$ dropped to ~10 mM close to the soma, while it declined to ~13 mM in most distal branches. Figure 8d (right) illustrates the relative change in $[Na^+]_p$ and peak amplitude with respect to soma for the longest branch with various sub-branches.

Taken together, these simulations demonstrate that the specific subunit composition of the NKA, together with its expression level and the magnitude of $Na^+$ influx, are important determinants of the

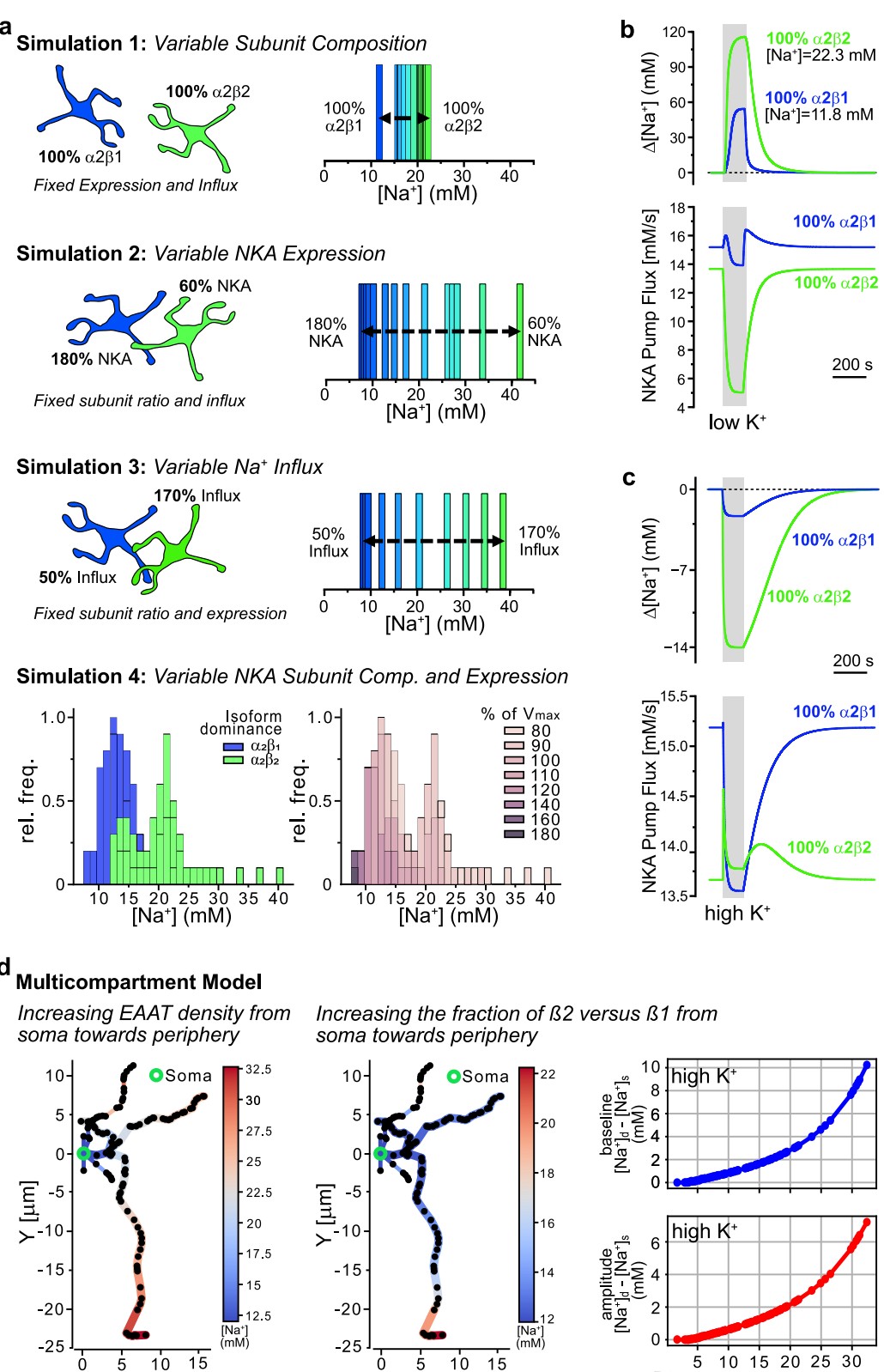

baseline [Na$^+$] of astrocytes. Moreover, they show that a combination of these parameters can reproduce the relatively broad, biphasic distribution of astrocyte [Na$^+$] determined experimentally. In addition, the simulations replicate the experimentally observed increase in [Na$^+$]$_p$ with increasing distance from the soma as well as the dependence of the magnitude of [Na$^+$] changes induced by inhibition of the NKA and its stimulation by increases in extracellular [K$^+$]$_e$.

## Discussion

Employing MP-FLIM, we performed a quantitative analysis of the intracellular [Na$^+$] of astrocytes in mouse forebrain tissue slices and cortex in vivo[24]. Our measurements revealed a mean somatic [Na$^+$] of 13-14 mM in astrocytes of juvenile animals (P14−20) in situ, which is similar to values reported using intensity-based imaging[26,36−40]. At P14−20, several major steps in functional astrocyte maturation are

**Fig. 8 | Biophysical modeling of astrocytic [Na$^+$]. a** Results from simulations illustrating the determinants of [Na$^+$]. 1: Influence of varying NKA subunit composition between 100% α2β1 to 100% α2β2 on [Na$^+$], while NKA expression levels/pump strength and Na$^+$ influx are constant. 2: Variation of NKA expression from 180-60% at fixed Na$^+$ influx and NKA subunit composition (α2β1/α2β2 = 30/70). 3: Variation of Na$^+$ influx from 50–170% at fixed NKA subunit composition (α2β1/α2β2 = 30/70) and fixed NKA expression (100%). 4: Simulation combining variable isoform dominance with variable NKA expression as indicated. Left: Bar-chart illustrating whether α2β1 (blue) or α2β2 (green) dominate. Right: Same as left, but color-code indicating relative NKA expression (V$_{max}$). **b** Top: Simulation showing [Na$^+$] changes induced by K$^+$-free saline for 2 minutes ("low K$^+$"; gray shaded area) in astrocytes with exclusive expression of α2β1 (blue) or α2β2 (green). Bottom: Time traces of flux through the two NKA compositions during low K$^+$-simulation (α2β1:

blue, α2β2: green). **c** Same as in b, but showing [Na$^+$] changes induced by an increase in [K$^+$]$_e$ to 10 mM ("high K$^+$"; gray shaded area) (top) and time traces of NKA flux during high K$^+$ (bottom). **d** Left: Multicompartment simulation showing [Na$^+$] changes with variation in EAAT density from 100–300% as a function of distance from soma at fixed peak NKA expression (100%) and composition (α2β1/α2β2 = 30/70). Center: Multicompartment simulation showing [Na$^+$] changes with variation in the subunit expression ratio as a function of distance from soma (α2β1 dominant near soma and α2β2 dominant in the processes) at fixed NKA expression and EAAT density (100%). Right: Change in baseline [Na$^+$] (upper) and peak amplitude [Na$^+$] (lower) relative to the soma as functions of distance from soma for the longest branch with different sub-branches in response to high K$^+$ stimulation. Subscripts s and d represent soma and distance from the soma. Source data are provided as a Source Data file.

terminated[41–44], and glutamate-induced Na$^+$ transients are essentially indistinguishable to those of 2-3 months old animals[38]. However, further changes in expression levels of Na$^+$-dependent transporters may influence astrocytic [Na$^+$] in adults and during aging.

While reliable calibrations cannot be performed in vivo, our data indicate that the somatic [Na$^+$] of cortical astrocytes in vivo is several mM higher. Of note, the FL of ING-2 is largely independent from temperature[24], indicating that this difference is not due to the higher temperature of recordings performed in vivo. A plausible explanation is that higher levels of neuronal activity in vivo lead to increased neurotransmitter release and higher astrocytic neurotransmitter uptake along with Na$^+$, compared to brain slices. However, our in vivo measurements were performed under isoflurane anesthesia and with buprenorphine analgesia, which can profoundly affect brain function and homeostasis (e.g., gas exchange and metabolic state). For instance, isoflurane anesthesia is well-known for its effects on neuronal firing rates and brain circulation and its inhibition of astrocytic Ca$^{2+}$ signals[45]. Regarding astrocytic Na$^+$ levels, two other observations could be relevant. Deep isoflurane anesthesia has been shown to reduce NKA activity by ~25%[46], which could increase the steady-state [Na$^+$]. Also, isoflurane was shown to increase glutamate uptake by cultured astrocytes[47], which may also explain an elevated [Na$^+$]. Such potential effects of anesthesia could be unmasked in the future by recordings in awake mice using genetically encoded Na$^+$ indicators, which are not available at the moment.

Both in situ and in vivo, somatic [Na$^+$] exhibited a relatively broad range. Na$^+$ is a highly mobile, unbuffered ion, suggesting its equilibration within astrocytes and between gap-junction-coupled cells at rest[12,48]. Here, we found that the variance of [Na$^+$]$_s$ was smallest for somata within 50 μm of each other, and that inhibition of gap junctions with carbenoxolone increased the overall range of [Na$^+$]$_s$, indicating that astrocytic [Na$^+$]$_s$ is at least partly determined by diffusion-driven exchange of Na$^+$ between cells as reported in culture[48]. Moreover, carbenoxolone caused an increase in the average [Na$^+$]$_s$, which could be related to increased neuronal excitability, as described in mice deficient for astroglial Cx30 and Cx43[49]. This is in line with the slight increase in [K$^+$]$_e$ induced by carbenoxolone, also reported from astrocyte-specific Cx43/Cx30-deficient mice[31]. However, unspecific effects cannot be ruled out as carbenoxolone has also direct effects on neurons[50,51].

FL-based imaging of a large number of cells revealed that in hippocampal slices, astrocytic [Na$^+$]$_s$ follows a bimodal distribution with two Gaussian components that peak around 9 and 17 mM. A similar biphasic distribution has been described for basal [Ca$^{2+}$] of hippocampal astrocytes in situ and of cortical astrocytes in vivo[22]. These observations suggest the existence of at least two distinct functional subgroups, supported by the reported molecular heterogeneity of astrocytes[27,29,30,52–54]. Notably, this molecular heterogeneity includes NKA isoforms. Rodent forebrain astrocytes predominantly express the α2 isoform, which is combined with β1 in approximately 10–30% of cells, while the remainder expresses β2[27,55]. Our own results are consistent with these findings, as both mRNA and protein staining patterns

aligned with the reported lower levels of β1 compared to β2. Furthermore, they suggest a higher overall β2 expression on astrocyte processes than β1. Based on their different ion binding affinities[4,55], these two NKA subunit compositions (α2ß1 versus α2ß2) are associated with differences in baseline [Na$^+$], as also demonstrated by our simulations.

The experimentally observed heterogeneity in astrocytic [Na$^+$] is thus likely to be due, at least in part, to different NKA isoforms, with exclusive α2β1 expression resulting in low and α2β2 in high baseline [Na$^+$]. Altering the overall NKA expression in our simulations from 60–180% caused this range to increase even further, indicating that different expression levels can also contribute to the broad distribution of basal [Na$^+$]. Finally, both experiments and simulations strongly suggest that differences in Na$^+$ influx rates influence [Na$^+$]. Following NKA inhibition, cells with a high baseline [Na$^+$] (α2β2) exhibited a larger [Na$^+$] increase than those with a low baseline [Na$^+$] (α2β1), indicating stronger Na$^+$ influx into the former.

Moreover, our results revealed substantial subcellular heterogeneities. [Na$^+$]$_p$ not only varied between different main processes of a given cell, but also increased with increasing distance from the soma. These results indicate that astrocyte processes regulate [Na$^+$] independently from the soma, a finding similar to what was described for astrocyte [Ca$^{2+}$][22] and [Cl$^-$][56]. This is surprising given the high mobility of Na$^+$ and the absence of Na$^+$-buffer systems[7], but in line with modeling studies suggesting the existence of specific intracellular Na$^+$ domains, caused by the retention of cations in fine astrocyte processes[14,15].

Our experiments also suggest that EAATs mediate small, but detectable Na$^+$ influx in unstimulated conditions, as their inhibition with TFB-TBOA caused astrocytic [Na$^+$]$_s$ to decline. The magnitude of this decline depended on the initial [Na$^+$], as high [Na$^+$]-astrocytes showed a greater fall in [Na$^+$] than those with a low initial [Na$^+$]. In this context, it is noteworthy that the [K$^+$]$_e$ increase induced by TFB-TBOA was only a fraction of that seen with high K$^+$ perfusion, suggesting that the observed effects were at least partly attributable to decreased Na$^+$ influx following EAAT inhibition. Na$^+$ uptake by EAATs is also a main mechanism responsible for transient increases in astrocytic [Na$^+$]$_s$ upon glutamatergic neuronal activity in physiological and pathophysiological conditions[3,7,9,10]. EAATs are predominantly expressed in astrocyte processes and their surface density increases from the soma towards their tips[57–61]. The observed increase in [Na$^+$]$_p$ from the soma towards the periphery may be thus generated, at least partially, by increased entry of Na$^+$ into distal processes via EAATs compared to proximal processes, a scenario supported by our simulations. In addition, the variable proximity of astrocyte processes to glutamate-releasing synapses will contribute to the heterogeneity in [Na$^+$]$_p$[17,18]. EAATs are co-localized with α2-NKA in astrocyte processes, which is most highly expressed close to synapses[57,58,62]. Intriguingly, the Na$^+$ gradient along processes, as demonstrated here, will enable the directed diffusion of Na$^+$ from the periphery towards the large-volume soma, reducing local ATP consumption in processes by NKA-related export of Na$^+$[12,13,63].

We also found that the baseline $[Na^+]$ of a given compartment determined its decline in response to increases in $[K^+]_e$, which are mainly cleared by the NKA[4]. It is interesting to note that NKA and other $Na^+$-dependent transporters are temperature-dependent. For instance, decay constants for recovery from global intracellular $Na^+$ transients decreased by a factor of about 1.7 when increasing the temperature by $10\,°C$[64,65], exemplifying the dominant role of NKA in restoring the electrochemical $Na^+$ gradient[66]. Therefore, the kinetics of $K^+$-induced activation of NKA and the resulting decline in $[Na^+]$ are expected to be faster at physiological than at room temperature.

High-baseline $[Na^+]$ astrocytes showed a stronger $[Na^+]$ decline upon $[K^+]_e$ elevation than low-baseline $[Na^+]$ astrocytes, suggesting that the former undergo a stronger $K^+$-induced activation of the NKA than the latter. Due to its lower affinity for extracellular $K^+$ ($K_m$- 3.6 mM), the α2β2 combination is specifically geared towards uptake of $K^+$. In contrast, α2β1 has lower affinity for intracellular $Na^+$ ($K_m$- 10.6 mM) and is thus more efficiently activated by $[Na^+]$ increases[4,55]. Indeed, our simulations demonstrate that the NKA-pump current increased in α2β1 astrocytes in response to $Na^+$ loading, this was not the case in α2β2 cells. Moreover, α2β1 astrocytes showed a much smaller decrease in $[Na^+]$ in response to increases in $[K^+]_e$ than α2β2 astrocytes. Along this line, increasing the fraction of ß2 versus ß1 expression in distal processes also resulted in a higher baseline $[Na^+]_p$ and a stronger $[K^+]_e$-induced decline. This is consistent with our experimental results demonstrating that the magnitude of the $K^+$-induced $[Na^+]$ decrease and NKA activation, respectively, was positively correlated with the initial baseline $[Na^+]$. This again clearly indicates that low-baseline $[Na^+]$ astrocytes and compartments predominately express α2β1, whereas high-baseline $[Na^+]$ astrocytes and compartments predominately express α2β2.

In conclusion, our study reveals a substantial cellular and subcellular heterogeneity in astrocyte baseline $[Na^+]$. We demonstrate that the broad range of astrocytic $[Na^+]$ can be explained by the combination of specific NKA isoforms (α2β2 versus α2β1) and expression levels, as well as by differences in the $Na^+$ influx via transporters such as EAATs. Notably, these players also determine the strength of NKA activation and of uptake of $K^+$ and glutamate from the ECS. Our results also show that astrocytic compartments which display a high $[Na^+]$, indicative of expression of α2β2 and a high rate of $Na^+$ influx via EAATs and other $Na^+$-dependent transporters, are especially geared towards an efficient clearance and control of extracellular ion and neurotransmitter homeostasis, thereby shaping and controlling the excitability of surrounding neurons and networks.

## Methods

### Animals

The study was conducted in accordance with all relevant ethical regulations, and all experiments received ethical approval if necessary. All relevant national and institutional guidelines and requirements, as well as the guidelines of the Heinrich Heine University Düsseldorf (HHU) and the European Community Council Directives (2010/63/EU and 86/609/EEC), were strictly obeyed. Mice were housed at a room temperature of 20–22° Celsius and a humidity level of $55\pm5\%$ under 12 h light/dark conditions with food and water access *ad libitum*. Experiments using tissue slices were approved by the Animal Welfare Office at the Animal Care and Use Facility of the HHU (Institutional Act No. O52/05) and the Government of Middle Franconia (No. 55.2.2-2532-2-1322 and TS20/2020). For in vivo experiments, all procedures were approved by the Landesamt für Natur, Umwelt und Verbraucherschutz Nordrhein-Westfalen (No. 81-02.04.2018.A330; LANUV, Germany), where required.

### Experimental design and statistical analysis of physiological experiments

Each set of experiments was performed on at least three different animals. Sex was not considered, and data were not disaggregated for

sex, as there is no evidence that sex has an effect on the cellular parameters examined here. In situ experiments were performed on at least 4 different slices obtained from at least 3 different mice of both sexes. In vivo imaging was performed on 3 male mice. If not stated otherwise, $n_p$ is the number of individual processes analyzed, $n$ represents the number of analyzed cells, $N$ represents the number of slice preparations in situ.

The data was tested for normal distribution using the Shapiro-Wilk test. Normally distributed paired data was statistically analyzed by a paired-t-test. Unpaired parametric data was analyzed with a two-sample-t-test for two groups, and with a one-sided ANOVA with Bonferoni post-hoc correction for more than two groups. Not-normally distributed paired data was analyzed with a Wilcoxon-signed-rank-test, unpaired data with a Mann-Whitney-test. All tests were two-sided. Mixed effects linear regression analysis, in which individual mice and slices were treated as random effects and the $[Na^+]$ as a fixed effect, was performed in Python using the statsmodels module (version 0.14.6). To evaluate the quality of the compared models (e.g., whether single or double Gaussian fits were appropriate) the F-test, $\Delta Chi^2$ test and Akaike information criterion were applied using Origin Pro2025 (OriginLab Corporation, Northampton, USA). The results of the Akaike information criterion are given as "1 G" for single and "2 G" for double-Gaussian fits. For a detailed list of the statistical tests performed and their outcomes, please refer to the Supplemental Statistical Summary. The results of the tests are illustrated as follows: *: $0.01 \leq p < 0.05$, **: $0.001 \leq p < 0.01$ and ***: $p < 0.001$.

### Preparation, salines and drug application

Unless otherwise specified, tissue slices were prepared from BALB/c mice (both sexes), bred and raised by the Animal Welfare Office at the Animal Care and Use Facility of the HHU, using standard procedures[67]. In brief, mice of postnatal days (P)14–20 were anesthetized with $CO_2$, rapidly decapitated, and their brains removed. Hemispheres were cut into $250\,\mu m$-thick slices in ice-cold preparation artificial cerebrospinal fluid (pACSF), containing (in mM): 130 NaCl, 2.5 KCl, 0.5 $CaCl_2$, 6 $MgCl_2$, 1.25 $NaH_2PO_4$, 26 $NaHCO_3$, and 10 glucose, bubbled with 95% $O_2$/5% $CO_2$, pH 7.4, osmolarity 310 mOsm/l. Afterwards, slices were incubated at $34\,°C$ for 20 min in pACSF containing $0.5$-$1\,\mu M$ sulforhodamine 101 (SR101) to stain astrocytes. Subsequently, they were transferred to standard ACSF, containing (in mM): 130 NaCl, 2.5 KCl, 2 $CaCl_2$, 1 $MgCl_2$, 1.25 $NaH_2PO_4$, 26 $NaHCO_3$, and 10 glucose, bubbled with 95% $O_2$/5% $CO_2$, pH 7.4, osmolarity 310 mOsm/l for 10 minutes at $34\,°C$, after which slices were kept in ACSF at room temperature ($21\pm1\,°C$).

During experiments, slices were continuously perfused with ACSF at room temperature. To alter $[K^+]_e$, slices were perfused with ACSF in which the $[K^+]$ was adjusted to 0 or 10 mM while keeping $[Na^+]+[K^+]$ at 160 mM. Pharmacological blockers (tetrodotoxin (TTX, HLB-HB1035, Biozol, Hamburg, Germany); carbenoxolone (CBX, C4790-5G, Sigma Aldrich); (2S,3S)-3-[3-[4-(trifluoromethyl)-benzoy-lamino]benzyloxy]aspartate (TFB-TBOA, CAS: 205309-81-5, Tocris, Bio-Techne, MN, USA); 6-cyano-7-nitroquinoxaline-2,3-dione (CNQX, CAS: 115066-14-3, Cayman Chemical, Biomol, Hamburg, Germany); DL-2-Amino-5-Phosphonovaleric acid (DL-AP5, CAS: 76326-31-3, Hello Bio, Dunshaughlin, Ireland) were added to the ACSF and applied by bath perfusion.

For calibration of ING-2 FL, slices were perfused with calibration salines containing (in mM): 10 HEPES, 16-136 K-gluconate, 0–150 $Na^+$, 0–150 $K^+$ (total concentration of NaCl+KCl: 150 mM), pH adjusted to 7.4 with KOH. In addition, calibration salines contained the $Na^+$ channel-forming antibiotic gramicidin ($3\,\mu M$), monensin ($Na^+$/$H^+$ antiporter; $10\,\mu M$) and ouabain (NKA inhibitor; $100\,\mu M$) (Calbiochem, Merck KGaA Darmstadt, Germany) to permeabilize cellular plasma membranes for $Na^+$[26,68].

## Fluorescence lifetime imaging in situ

For Na$^+$ imaging, the membrane-permeable form of ION-NaTRIUM-Green-2-AM (ING-2; Mobitec GmbH, Göttingen, Germany, #2011F) was pressure-injected as reported before[24]. Alternatively, the membrane-impermeable form of ING-2 was loaded via a patch-pipette (see below). Multi-photon fluorescence lifetime imaging microscopy (MP-FLIM) of ING-2 was performed using a modified laser-scanning microscope based on an A1-R MP system (Nikon Europe, Amsterdam, The Netherlands), equipped with a water immersion objective (NIR Apo 60x/NA 1.0, Nikon). Laser pulses ( < 100 fs, 840 nm) were generated at 80 MHz by a mode-locked Titan Sapphire laser (Mai Tai DeepSee, Newport, Spectra Physics; Irvine, CA, USA).

Images were acquired at ~ 1 Hz and temporally binned depending on the required temporal resolution. Average fluorescence lifetimes (FL) were measured using time-correlated single photon counting (TCSPC) with a spatial resolution of 0.41 x 0.41 μm per pixel. Fluorescence emission was split with a 560 nm long-pass dichroic mirror (H 560 LPXR, F48-562 AHF Analysentechnik AG, Tübingen, Germany) and band-pass filtered at 540/25 nm (F34-540A, AHF) for ING-2 and at 640/20 nm (F39-641, AHF) for SR101, before being directed to PMA hybrid photodetectors (PicoQuant, Berlin, Germany). TCSPC electronics (Multiharp 150, PicoQuant) and acquisition software (SymPhoTime64, Version 2.6, PicoQuant) were used for obtaining FL at a pixel dwell time of 3.81 μs for frames of 512 x 512 pixels. Astrocytes were selected using the SR101 image (Supplementary Fig. 4) and only regions of interest (ROIs) with a total photon count of > 2000 photons and at least 5 photons/pixel/frame were analyzed (Supplementary Fig. 3). Imaging parameters and analysis criteria were established in initial exploratory experiments for measuring larger populations of astrocytes repeatedly. Acquired images were analyzed using Symphotime64 software (Version 2.9, PicoQuant).

The amplitude-weighted average decay constant $\tau_{AVG}$ was calculated using the rapidReconvolution algorithm[24]. For calibration of ING-2 FL, the relationship between [Na$^+$] and $\tau_{AVG}$ was approximated by a shifted Michaelis-Menten kinetics function $\tau = (\tau_{max}*[Na^+])/(K_m + [Na^+]) + \tau_{min}$, where $\tau_{min}$ corresponds to [Na$^+$] = 0 mM and $\tau_{max}$ as well as $K_m$ are computed from the fitted function using OriginPro2025. Average FL data from in situ experiments was constrained to the boundaries of the calibration and dynamic range of the dye, respectively (lower limit: 0 mM, upper limit: 100 mM Na$^+$, compare Fig. 1a–e).

In addition, we tested for potential Na$^+$-independent changes in FL induced by various drugs by monitoring the FL of SR101. Supplementary Fig. 5 illustrates that SR101-FL was independent from Na$^+$ as it did not change with very large increases in Na$^+$. Moreover, it was not affected by the application of TTX, CBX, nor TFB-TBOA.

3D-distance calculations and correlation with [Na$^+$]$_i$ between astrocytes (compare Fig. 4f–h) were performed using a custom-written ImageJ routine on z-stacks of acute slices stained with ING-2 and SR101. Swelling of neuronal cell bodies following inhibition of EAATs by TFB-TBOA was analyzed based on intensity images derived from FL datasets and automated area analysis as described in Supplementary Fig. 7.

## In vivo MP-FLIM and analysis

Acute in vivo MP-FLIM was performed in layer 2/3 of the barrel cortex in postnatal week 8-12 C57BL/6 (Charles River Laboratories) male mice[23,69,70]. Mice were anesthetized using isoflurane inhalation in medical oxygen (3.5% for induction, 1.0–1.5% for maintenance, ~ 0.5 L/min) and additional buprenorphine analgesia (0.1 mg/kg). The mouse was head-fixed in a stereotaxic apparatus, the skull exposed, and a small craniotomy (3 mm diameter, 1.5 mm posterior to bregma and 3.5 mm lateral from the midline) above the right barrel cortex was performed using a dental drill. The skull was carefully removed, and the dura mater was kept intact. Then, 300 nl of a solution containing 750 μM ING-2-AM, Pluronic-127F (10% in DMSO) and SR101 (25 μM) dissolved in ACSF composed of (in mM): 152 NaCl, 2.5 KCl, 2 CaCl$_2$, 10

HEPES, 1.25 NaH$_2$PO$_4$, 1 MgSO$_4$, 10 D-glucose, pH 7.4, was pressure-injected via a glass micropipette at a depth of 120–150 μm from the dura mater at a rate of 100 nl/min. The cortex was then covered with 1.2% low-melting agarose. A glass coverslip (4 mm diameter, Warner Instruments, Hamden, USA) was placed on top and secured with dental cement (Temdent Classic, Schütz Dental GmbH, Rosbach, Germany). A custom-made metal frame was attached to the skull using dental cement. The eyes were covered with Bepanthen (Bayer Vital GmbH, Leverkusen, Germany) throughout the entire surgical and imaging procedure.

For recordings, mice were head-fixed to another stereotaxic frame and placed under a two-photon excitation microscope (COSYS Ltd, East Sussex, UK) equipped with a femtosecond infrared pulsed MaiTai HP laser (Spectra-Physics) and a 16 × water-immersion objective (Nikon LWD, NA 0.8). The excitation wavelength was set to 840 nm. To prevent tissue damage, laser power was kept < 30 mW at the front lens of the objective. Images (512 × 512 pixels) were acquired at a depth of 100–150 μm from the pia with a nominal resolution of ~ 0.4 μm/pixel. Emitted fluorescence was spectrally separated using an appropriate dichroic mirror and filter set. For astrocyte identification, SR101 was imaged in intensity mode using ScanImage 2022.1.0 (MBF Bioscience, Williston, USA) (Supplementary Fig. 2), while ING-2 fluorescence (green channel) was monitored in parallel using TCSPC and a Multi-Harp 150 module (PicoQuant) connected to a cooled detector (PMA Hybrid 40, Picoquant). FLIM data was acquired with SymPhoTime64 software (PicoQuant). Images (20-100 frames) were acquired at a rate of 0.95 Hz. Recordings were performed under continued isoflurane inhalation anesthesia (1.0–1.5% in medical oxygen, ~ 0.5 L/min) to keep the breathing rate at 55–60 breaths/minute[71]. Animals were kept at a constant temperature of 37 °C throughout the surgery and the experiment by placing them on a heating blanket.

In vivo data were motion-corrected and analyzed offline using in lab-written MATLAB scripts (The MathWorks, Natick, USA) and OriginPro 2024b. ING-2 FLs were quantified for somatic ROIs and converted to [Na$^+$]$_s$. This analysis was performed as explained above, but with minor modifications. A separate calibration was performed in a cuvette on the in vivo microscope to account for technical differences (e.g., light source, excitation and emission paths, fluorescence detection). Because reliable intracellular calibration is not possible deep within the brain in vivo, we compared the decay time constants of individual components of in vitro (cuvette) calibrations and in vivo recordings and found small differences (in vitro: $\tau_1 = 0.49$ ns/$\tau_2 = 2.51$ ns; in vivo: $\tau_1 = 0.44$ ns/$\tau_2 = 2.59$ ns). We then estimated the effect of these differences on the calculated [Na$^+$]. For a set of 400 randomly chosen decay component amplitudes $A_1$ and $A_2$, we determined the resulting amplitude-weighted decay time constants $\tau_{avg,rec}$ and $\tau_{avg,cal}$ using the values for $\tau_1$ and $\tau_2$ from in vivo recordings and calibration, respectively. The relationship between $\tau_{avg,rec}$ and $\tau_{avg,cal}$ was linear ($\tau_{avg,rec} = -0.076 + \tau_{avg,cal} * 1.063$; $R^2 = 1$), suggesting a small effect of the in vivo environment on the relationship between [Na$^+$] and $\tau_{avg}$ of ING-2. Translating recorded $\tau_{avg,rec}$ into the corresponding $\tau_{avg,cal}$ using the above relationship before calculating [Na$^+$] lowered the average resulting [Na$^+$] by ~0.5 mM. Because this small deviation did not qualitatively affect our conclusions, we omitted any corrections and directly converted $\tau_{avg,rec}$ into [Na$^+$]. For further details about this approach and an experimental confirmation for Ca$^{2+}$-FLIM, see ref. 23. It should also be noted that the environment of ING-2 is likely to differ between recordings in vivo and in tissue slices (e.g., because of different pO$_2$, pCO$_2$, metabolic state), which may result in different ING-2 properties in both conditions.

## Image processing

All images presenting FL are fast lifetime images and were processed equally for illustration purposes regarding adjustment of brightness and contrast within each respective figure. To better visualize

astrocyte processes, the fast lifetime images were masked with SR101 fluorescence intensity as shown in Supplementary Fig. 4 and indicated in the respective figure legends. This effectively masks areas that appear dark in the SR101 intensity image, thereby increasing the visual perception of primary and secondary astrocyte processes.

For dynamic imaging of astrocytic processes, XYZT stacks of variable size (3–17 steps) were performed with a temporal binning of 1–10 frames for each step. Either XYZT stacks were summed in z-axes prior to the analysis of individual processes, or processes were tracked in XYZT for correct ROI placement.

## Widefield imaging of SBFI

For intensity-based ratiometric imaging of astrocyte $[Na^+]_s$[26], the membrane-permeable form of SBFI (SBFI-AM, sodium-binding benzofuran isophthalate acetoxymethyl [AM] ester, ION Biosciences, LLC., Texas, USA) was pressure-injected into the hippocampal CA1 region of acute tissue slices. Wide-field imaging was performed at an epifluorescence microscope (Nikon Eclipse FN 1, Nikon Europe, Düsseldorf, Germany), combined with a Fluor 40x/0.8 NA water immersion objective (Nikon), an orca FLASH 4.0 LT camera (Hamamatsu Photonics Deutschland GmbH, Herrsching, Germany), and a Poly-V monochromator TILL Photonics/FEI, Planegg, Germany). For ratiometric imaging, SBFI was alternately excited at 340 and 380 nm. Images were obtained at 1 Hz, and emission was collected between 468 and 552 nm. The SBFI-fluorescence ratio ($F_{340}/F_{380}$) was calculated for individual ROIs positioned around SR101-positive somata and analyzed using NIS-Elements software (Nikon) and OriginPro software (OriginLab Corporation, Northampton, MA, USA). For converting SBFI fluorescence ratio into $[Na^+]$, calibrations were performed by perfusing slices with calibration salines containing 0–150 mM $Na^+$, the ionophores gramicidin (3 μM) and monensin (10 μM), as well as the NKA inhibitor ouabain (100 μM)[26]. A Michaelis-Menten fit of the mean data points obtained for each $[Na^+]$ ($n = 138$-508, $N \geq 3$ for each $[Na^+]$) revealed an apparent $K_D$ of 35 mM ($R^2 = 0.996$) (Supplementary Fig. 1).

## Electrophysiology

Cells were subjected to patch-clamp using an EPC10 amplifier and PatchMaster NEXT or PatchMaster software (MCS GmbH/HEKA Elektronik, Reutlingen, Germany). Patch pipettes were pulled from borosilicate glass capillaries (GB150(F) 8 P, Science Products, Hofheim am Taunus, Germany) using a vertical puller (PC-10 Puller, Narishige International, London, UK). The pipette solution for whole-cell patch-clamp of astrocytes contained (in mM): 116 K-gluconate, 32 KCl, 10 HEPES (N-(2-hydroxyethyl)piperazine-N′-2-ethanesulfonic acid), 10 NaCl, 4 Mg-ATP, 0.4 $Na_3$-GTP and 0.05 ING-2 (MoBiTec; Göttingen, Germany), pH 7.3, osmolarity 305 mOsm/l. Cells were held in whole-cell mode for ≥15 min prior to starting imaging experiments. For cell-attached measurements of neurons, pipettes were filled with HEPES-buffered saline, containing (in mM): 125 NaCl, 3 KCl, 25 HEPES, 2 $MgSO_4$, 2 $CaCl_2$, 1.25 $NaH_2PO_4$ and 10 glucose, pH 7.4, osmolarity 315 mOsm/l. Data was analyzed using OriginPro2025.

Double-barreled ion-sensitive microelectrodes were employed for measurement of $[K^+]_e$[72]. In brief, two borosilicate glass capillaries with filament (GC100F-15, GC150F-15; Harvard Apparatus, Holliston, MA, USA) were glued and pulled out together. The $K^+$-sensitive barrel was filled with valinomycin (Ionophore I, Cocktail B, Merck), backfilled with 100 mM KCl. Reference channels were filled with HEPES-buffered saline (see above). Measurements were performed using Axoscope (version 8.1.0.07; Axon Instruments) and analyzed using Clampfit (Axon Instruments, pCLAMP Version 8.1, Molecular Devices, San Jose, U.S.A.). Electrodes were positioned in the *stratum radiatum* at ~50 μm below the slice surface, and electrodes were re-calibrated after each experiment. Calibration of $K^+$-sensitive microelectrodes was performed using salines composed of 25 mM HEPES and a total of 150 mM NaCl + KCl, in which $[K^+]$ was 0–10 mM and $[Na^+]$ adjusted accordingly.

## Immunohistochemistry

Animals were decapitated following $CO_2$ anesthesia. Brains were quickly removed, immersion-fixed for 1-2 days at 4 °C in 4% paraformaldehyde (PFA) in phosphate-buffered saline (PBS; pH 7.4) and stored in PBS for ≥1 day at 4 °C until sectioning. Parasagittal hippocampus sections (25 μm) were sliced for immunohistochemical processing on an HM650V Vibratome (Thermo Scientific, Limburg, Germany). All chemicals were purchased from Sigma-Aldrich (Munich, Germany) if not stated otherwise. Detailed information on the number of animals, sections and on antibodies employed is listed in Supplementary Tables 1-4.

To label for the NKA subunit ß1, sections were permeabilized and blocked in PBS containing 0.25% Triton-X-100 (TX) and 2% normal goat serum (NGS, Invitrogen; blocking reagent; 90 min, 4 °C) followed by incubation (overnight, 4 °C) with the primary antibody, 3 x 20 min washes in PBS/NGS/Triton (60 min, RT), incubation with the secondary antibody, 3x20 min washes in PBS and coverslipped with Mowiol/DABCO (Fluka; Calbiochem, San Diego, CA). The labeling for ß2 was conducted accordingly, but no TX was added. To identify astroglia or neurons, tissue subsets of NKA isoform-labeled sections were either double-labeled for glial acidic fibrillary protein (GFAP) or microtubule-associated protein-2 (MAP-2). For double-labeling of ß1 and ß2 in identical sections, tissue was first processed for ß2, coverslipped, documented, bleached for 3 h under the UV of a sterile hood, de-coverslipped, then labeled for ß1, finally coverslipped and documented. Negative controls were run in parallel to each staining by omitting either all or one of the primary antibodies. Control stainings omitting one of the primary antibodies exhibited identical labeling patterns for the remaining antibody as for the double stainings. Excluding both primary antibodies never resulted in a labeling product.

For documentation, a fully automated microscope (90i, Nikon Europe, Amsterdam) equipped with a Nikon 20x Apo-Plan VC and with NIS-Elements software was employed. Immunophotomicrographs were assessed utilizing maximum intensity projections of whole-frame widefield z-stacks (5 optical sections) or stitched z-stacks. Stacks were corrected for spherical aberration. For documentation reasons, images were corrected for brightness and contrast, merged in Adobe Photoshop and mounted in Adobe Illustrator.

## RNAscope combined with immunohistochemistry

To quantify mRNA expression, Advanced Cell Diagnostics (ACD) RNAscope® Multiplex Fluorescent V2 Assay was performed in combination with immunohistochemistry (IHC) according to the manufacturers protocol[73]. Two-weeks-old mice (3 animals, P 14, both sexes; 632C57BL/6 J, Charles River Laboratories) were killed in $CO_2$, followed by transcardial perfusion with phosphate-buffered saline (PBS; pH 7.4) for 2 min and then 4% paraformaldehyde (PFA) in PBS, and brains stored overnight in 4% PFA. Mouse brains were transferred into 30% sucrose solution before being embedded in Leica Tissue Freezing Medium and stored at − 80 °C. Mouse brains were sliced on a cryotome (Leica Microsystems, Wetzlar, Germany) to a thickness of 14 μm and stored at − 20 °C until RNAscope was performed on three slices of each animal. Initially, slices were dehydrated by baking in HybEZTM II in Bake Mode, followed by a standard series of ethanol treatment. Antigen retrieval was performed using ACD Co-Detection Target Retrieval Reagents at 99 °C. A hydrophobic barrier was drawn around the samples and maintained throughout the remainder of the protocol.

Primary antibodies for immunohistochemistry staining for GFAP (rabbit anti-GFAP; 1:500; Agilent, RRID: AB2811722) were applied and left overnight at 4 °C. Sections were prepared for probes and hybridization with Protease III followed by application of probes. Probes were designed and produced by ACD and included: Mm-Atp1b1-mRNA (cat. 531721: *Mus musculus* ATPase $Na^+$/$K^+$ transporting beta 1 polypeptide) in C1 channel and Mm-Atp1b2-transcript variant 1 mRNA (cat.

417131-C2: *Mus Musculus* ATPase, Na$^+$/K$^+$ transporting beta 2 polypeptide) in C2 channel. Opal dye 570 was conjugated to C1 probes and Opal dye 650 was conjugated to C2 probes, followed by secondary antibodies against GFAP (CF-488 anti-rabbit; Biotium), which were applied for 30 minutes at room temperature, followed by DAPI. Cover slips were allowed to dry overnight at 4 °C and imaging at the confocal microscope (Zeiss LSM 780; 63x objective). Astrocytes were identified by the presence of IHC staining for GFAP, and cell body location was confirmed with DAPI.

## Simulations

The equations and related parameters modeling the dynamics of membrane potential and ion concentrations in the astrocyte and ECS are described in the Supplementary Note 1 and Supplementary Tables 6-7. The rate equations were integrated using Euler method in Python 3.

## Reporting summary

Further information on research design is available in the Nature Portfolio Reporting Summary linked to this article.

## Data availability

The source data generated in this study are provided in the Source Data file. Further data are available from the lead contact upon request. Source data are provided in this paper.

## Code availability

The code used to compute the results and statistics, as well as generate visualizations of biophysical simulations, is deposited in a GitHub public repository at https://doi.org/10.5281/zenodo.19487413.

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

## Acknowledgements

This study was supported by the Deutsche Forschungsgemeinschaft (DFG, German Research Foundation: Project #461542557 and RO 2327/13-2,14-2 to C.R.R.; HE 6949/8 to C.H.; BE 5136/6-1 and BE5136/7-1 to R.B.), the Federal Ministry of Education and Research (BMBF), Germany (Project SynGluCross to C.R.R. and C.H.) and by the National Institutes of Health (R01NS130916 and R21AG087910 to G.U.). J.F. was supported by a scholarship (202306370039) from the China Scholarship Council, Beijing, China; N.C. was supported by a fellowship from the DFG-Research Training Group 2162 (Neurodevelopment and Vulnerability of the Central Nervous System). The authors wish to thank Claudia Roderigo and Simone Durry for expert technical assistance.

## Author contributions

Conceptualization, C.R.R., J.M. and G.U.; methodology, J.M., J.F., N.C., R.B., S.E., P.U., C.H., K.W.K., S.D. and V.B.; experiments, formal and biostatistical analysis: J.M., J.F., N.C., R.B., S.E., P.U., C.H. and V.B; biophysical modeling, A.B. and G.U.; writing – original draft, review and editing, all authors; supervision: C.R.R., C.H., R.B., and G.U.; funding acquisition, C.R.R., C.H., R.B. and G.U.

## Funding

## Competing interests

The authors declare no competing interests.
