## [Transparent Peer Review file · Nature Communications]

Cellular and subcellular heterogeneity of astrocytic Na⁺ homeostasis tuning astrocytes into functionally distinct subgroups in the mouse brain

Corresponding Author: Professor Christine Rose

Version 0:

Reviewer comments:

Reviewer #1

(Remarks to the Author)

Meyer et al. present a concise manuscript evaluating subcellular astrocytic Na⁺ concentration in vitro and in vivo, and explores how intracellular Na⁺ is influenced by a range of factors, most notable glutamate and K⁺ uptake, as well as Na/K-ATPase activity. They also present biophysical modelling results, which strengthen the manuscript, but which lies outside my expertise to evaluate.

This is a highly interesting and potentially very important study that significantly extends our understanding of astrocytic homeostatic functions. I think employing quantitative FLIM analyses as in the current study is essential to move the astrocyte physiology field forward. While overall I am highly enthusiastic about this study, I have some concerns, in particular about the interpretation of in vivo work and apparent lack of physiological monitoring during in vivo recordings that should – as a minimum – be addressed thoroughly by an elaborate discussion.

Comments and suggestions:

1. According to the In vivo microscopy methods section, only temperature was controlled for during imaging. If more extensive physiological monitoring was performed details must be provided. To be able to interpret physiological parameters from in vivo investigations of anesthetized mice, proper physiological monitoring is crucial, as mouse physiological parameters (especially in the aftermath of acute surgery) rapidly deteriorates. Isoflurane (even though in a relatively low dose was used here) combined with buprenorphine could profoundly affect mouse physiological parameters (as evident when evaluating arterial blood gas composition): Presumably the mice were slightly hypoxic unless oxygen was supplied with isoflurane (not stated), whereas if oxygen was supplied, they are quite likely to be hypercapnic, which again will have consequences for excitability, pH, metabolic state etc. Moreover, within a couple of hours system physiology rapidly deteriorates in mice during general anesthesia (in particular if the mouse has been subjected to acute surgery) and monitoring is of essence to ensure that results could be trusted.

Ideally physiological monitoring should be provided, but as this is primarily a slice study, as a minimum, a thorough discussion of these caveats is warranted.

2. The effects of anesthesia vs the unanesthetized state should be addressed when interpreting the in vivo data vs the in vitro data, and in the discussion. Performing this type of experiment in the unanesthetized state is of course very challenging if not impossible, but even so the interpretation of results must take the direct effects of anesthesia on brain state and the processes studied into account. One would expect that the drugs administered (iso/buprenorphine) would influence both neural activity and neuromodulator activity in dose and drug-dependent ways, in addition to potentially directly affecting ionic pump activity and channels. Hence, the explanation provided of the higher Na⁺ levels in the in vivo situation (“higher levels of neuronal activity”) should be expanded and elaborated to address all these aspects.

3. Slice experiments were performed at room temperature. The effect of temperature on Na⁺ levels (or rather the known effects of temperature on the Na/K-ATPase, GLT1 etc) should be appropriately discussed.

4. The in vivo surgical preparation procedure should be present in the methods section, not only references to previous

work.

5. Slice experiments were performed in mouse pups (likely for good reason to allow easier patch clamp), however this should be discussed when interpreting the results as there are known developmental changes in everything from resting membrane potential, Na/K-ATPase and GLT1 expression etc that could likely influence Na⁺ levels.

6. As far as I understand, only processes clearly visible by SR101 labelling were analyzed. However, astrocytes have processes that are considerably smaller than these, that cannot be resolved with the optical imaging employed. However, would it not still be possible to assess fluorescence lifetime in regions where these processes reside ("gliopil") even though they only occupy a fraction of the volume? Or is the sensor not properly filling these smallest of processes?

7. Even with the elaborate supplementary statistical summary it is somewhat difficult to understand which n/N has been used in the various comparisons. An example to illustrate: in Figure 6m, were recordings of cells directly compared (i.e. n = number of cells) or was the data binned (per slice/mouse or similar)? A potentially more sound approach that strikes a better balance between being too conservative (comparing means per slice) vs too likely to detect significant differences (n = observations) would be to employ a method that takes into account the hierarchical structure of the data where there are many observations per slice, and several slices per mouse (e.g. mixed effects linear regression models).

Reviewer #2

(Remarks to the Author)

Nature Communications manuscript NCOMMS-25-60820

"Cellular and subcellular heterogeneity of astrocytic Na⁺ homeostasis tuning astrocytes into functionally distinct subgroups" by Meyer et al.

The manuscript by Meyer et al describes the use of 2-photon fluorescence lifetime imaging for assessing astrocytic Na⁺ in acute brain slices and in vivo. To perform their study, the authors used the commercially available indicator ING-2. Using a pulsed infrared Ti:Sapphire laser coupled to time-resolved detectors, they detected and calibrated intracellular Na⁺ concentrations in astrocytes in situ. Of note, the same group previously published the detailed use of this approach in organotypic slice cultures (Meyer et al 2022) applied to neuronal Na⁺ assessment.

The presented approach convincingly demonstrates its ability to accurately monitoring astrocytic Na⁺ concentrations, and classical manipulations of intracellular Na⁺ (e.g. using low or high bath K⁺) led to expected changes. Using their approach, the authors found that astrocytes do not present a uniform Na⁺, contrary to previous results obtained from cultured astrocytes, notably by the same group. A bimodal distribution of resting Na⁺ was found in the population of astrocytes, interpreted by the authors as being likely linked to their differential expression of sodium pump isoforms (i.e. $\alpha 2\beta 1$ vs $\alpha 2\beta 2$), as supported by their mathematical modeling. They also found Na⁺ to be higher in astrocyte processes compared to somata, and that manipulations of sodium pump activity by external K⁺ has more pronounced effects in processes.

The study provides valuable insight into astrocytic Na⁺ in situ and in vivo. It must be noted that very few attempts of quantitative assessments of ion concentrations in cells within the tissue using imaging approaches have been reliably developed. The proposed approach using FLIM, rather than intensity-based signal detection, appears to robustly achieve this goal.

Arguably, the study seems to provide mainly incremental knowledge to the field and to a some extent confirms reasonable expectations. The potential role of different subunit compositions of the sodium pump in adjusting resting Na⁺ levels differentially in individual cells is sound and supported by the presented simulations. However, the lack of direct experimental validation limits the strength of this conclusion and leaves the central hypothesis insufficiently substantiated.

The paper is well written and interesting to read. This technically challenging study was carefully performed and provides a detailed report on use of FLIM for assessing intracellular Na⁺ of astrocytes in situ or in vivo.

Comments:

- Local syncytium: to what extent the locally connected astrocytes by gap-junction would actually share common Na⁺ levels? In other words, it is not clear if astrocytes with different Na⁺ levels observed in the study are not those more distant to each other or belonging to different syncytia. Previous studies have shown astrocytes to be extensively connected by gap junctions, a cellular organization for instance central to the regulation of interstitial K⁺ (potassium buffering). Is Na⁺ differently able to diffuse from cell to cell than other ions or small dye molecules?

- Local neuronal activity: individual astrocytes are embedded into local microcircuits, i.e. possibly surrounded by varying levels of electrical activity of neurons differentially releasing K⁺ or neurotransmitters. Could this explain different Na⁺ levels found in single astrocytes? It is for instance not clear whether the heterogeneity or bimodal distribution of Na⁺ levels persists in the when silencing neuronal activity (TTX).

- Differential expression of sodium pump isoforms: biophysical modelling presented and discussed by the authors supports a

direct effect of isoform composition on Na⁺ levels of individual astrocytes. To strengthen this hypothesis, the authors should provide direct assessment at the level of individual cells, of the existence of differential subunit composition, backed up e.g. by immuno stainings or spatial omics approaches, and ideally connect these results to the observed Na⁺ level in those cells. Such experiments could also address the question of whether the pump isoform composition differs between somata and processes, which could contribute to the observed different Na⁺ levels between the two compartments.

- p. 6, line 1: Effects of lowering external Na⁺: It would be worth mentioning how long it takes for cells to reach this lowest level after the switch to zero external Na⁺.

- p. 6, line 9: The study in vivo was performed with anesthetized head-fixed mice. Anesthesia is known to have a strong effects on astrocyte Ca²⁺ signaling compared to awake head-fixed mice. Do the authors have information whether this also applies to Na⁺ signaling or can the authors comment on whether such effect is expected as well for Na⁺?

- p. 9, starting last paragraph: Glutamate transporter inhibition and baseline Na⁺ levels:

The reviewer fails to see what the experiments using TBOA bring to the question of baseline Na⁺ values of astrocytes. It is for instance not described whether TBOA application influences the heterogeneity and bimodal distribution of intracellular Na⁺ in astrocytes.

Swelling of neuronal cell bodies after TBOA application: this effects most likely relates to glutamate excitotoxicity, as probably also assumed by the authors. Can the huge Na⁺ increase found in neurons (Fig 5) be prevented by blockade of glutamate receptor blocker? How to interpret those experiments in the context of the initial question of the determinants of baseline Na⁺ in astrocytes?

TBOA is a valuable non-transported competitive inhibitor of glutamate transporters. However, its affinity for EAATs is relatively low, compared to e.g. TFB-TBOA. The authors used 1 micromolar TBOA, a concentration much lower than the published EC₅₀ values, and to the one the same authors used previously (200 micromolar). The choice of this low concentration should be discussed.

- In vivo astrocytic cytosolic Na⁺: the authors write that calibration of ING-2 lifetimes for in vivo experiments was performed in a cuvette using calibration solutions without ionophores. While they claim that minor differences between fluorescence lifetime components observed in the cuvette and in vivo (p.22, lines 12-15), in their previous study using FLIM for neuronal Na⁺ (Meyer, J Neurosci 2022), they reported an apparent KD of 8.9mM in cuvettes, while in situ calibration in the neuronal cytosol yielded an apparent KD of >33mM, concluding that that the apparent KD of chemical sodium indicators is significantly higher inside cells as compared with cuvette calibration condition. If the same applied to the astrocyte Na⁺ of the present study, it could mean that the apparent higher Na⁺ levels claimed to be present in the in vivo brain compared to slices is due to this difference in KD. The authors should address this point and the validity of the absolute in vivo Na⁺ concentrations values presented.

Reviewer #3

(Remarks to the Author)

This manuscript uses exciting new imaging technologies to quantify astrocyte intracellular Na in situ and in vivo. This is very important work that provides information lacking in the field, as very little is known about the intracellular concentration of many ions in astrocytes. The work is highly rigorous, technically sound, and innovative. I am very enthusiastic about this study and am confident it will have a strong impact on the field. There are a number of moderate/minor concerns, in particular with how Na concentration was calculated and validated, that would greatly increase the acceptance of the quantifications as valid and accurate.

- The results say that the in vitro calibration curve was adjusted for 2P in vivo and is described in the methods. I did not see that section in the methods. Please add/update to include a description of how in vivo [Na⁺] was calculated.
- The most exciting aspect of this study is the intracellular quantification of astrocyte intracellular [Na⁺], so ensuring this calibration and calculation is correct is critical. The study would be strengthened by using neuron AP peaks, which can be calculated based on equilibrium potentials, as an independent validation that the [Na⁺] calculations in astrocytes are correct.
- Does ING-2 load astrocytes in vivo? In the 2P image shown, ING-2 looks very diffuse. Where is the dye located in this prep? The study would benefit from a reconstruction of an SR101 or genetically labeled astrocyte in vivo demonstrating ING-2 loading.
- If the intracellular solution dialyzes Na in the soma, shouldn't it do the same in the processes? Pipette withdrawal addresses this nicely, but is still difficult to understand. The paper cites a study showing similar findings in neurons, but references a pre-print from the Rose lab. Is there other data supporting this approach?
- The change in extracellular K in response to carboxoxelone is so tiny that its relevance is questionable. On the other hand, the authors report that TBOA increases K(e) by 2.5 mM, a highly relevant increase. Is this a statistically significant increase? Reporting these 2 changes should be contextualized better.
- How does low K⁺ drive neuronal activity to make increases in Na(S) affected by TTX? I would expect low K⁺ to decrease neuronal activity.
- What is the mechanisms by which TBOA causes such significant increases in neuron Na⁺ and swelling? Increased neuronal activity? Is this effect TTX dependent also?
- How do other manipulations tested (low K⁺, carboxoxelone, TBOA) affect Na in the processes of astrocytes? Many manipulations only report Na changes in the soma.

- Do the authors postulate that intracellular Na⁺ is the limiting factor in ATPase activity – when Na⁺ goes below a threshold does ATPase activity decrease/stop?

Minor concerns:

- How does perfusion of Na⁺ solutions onto slices lead to equilibrated intracellular Na⁺? The methods describe the use of gramicidin but this should be noted in the results section, at least briefly.
- The paper uses an AM version of ING-2, but this is only mentioned in the methods. Again, this should be mentioned in the results at least briefly.

Reviewer #4

(Remarks to the Author)

The manuscript describes the use of fluorescence lifetime-based imaging of astrocytes to determine the basal levels, compartmentalization, and regulation of Na⁺ levels, using both brain slice preparations and in vivo imaging in the mouse cortex. The authors use the unique ability of lifetime imaging to quantify and compare absolute Na⁺ levels, their response to pharmacological perturbations, and their cellular and subcellular heterogeneity. Their findings suggest the existence of significant variation in Na⁺ levels—particularly within astrocytic processes and branches—which has not been well characterized previously. The presence of small Na⁺ signaling domains in astrocytic processes may support critical local mechanisms, such as the modulation of glutamate spillover and synaptic remodeling.

Currently, there are technical limitations in addressing these questions. No genetically encoded Na⁺ indicators with sufficient signal-to-noise ratio are available to monitor Na⁺ levels. Furthermore, as the authors point out, intensity-dependent dyes present major challenges for quantifying absolute analyte concentrations, especially in the fine microdomains of astrocytic processes. To overcome this, the authors employ a Na⁺-sensitive dye and introduce an experimental approach to accurately quantify physiological Na⁺ levels in astrocytes. Two-photon fluorescence lifetime imaging microscopy (2pFLIM) is a particularly powerful and appropriate technique for this application.

Overall, the authors present valuable information in a clear and concise manner. I have a few comments and suggestions that could improve the rigor and impact of the study:

1. Calibration (Fig. 1): The authors show a dose-response curve to calibrate fluorescence lifetime against known Na⁺ concentrations, which forms the basis for their quantification of basal Na⁺ levels in astrocyte somata. They report heterogeneity among cells, with median Na⁺ levels falling toward the lower end of their calibration curve. Although the physiological values appear to fall within the linear range of the dye, calibration is based on only 1–2 data points in this critical range. Given the ~0.2 ns lifetime change observed here, more careful calibration is warranted to ensure accurate conversion from lifetime to Na⁺ concentration.
2. Slice vs. in vivo comparisons: The authors compare Na⁺ levels in cortical and hippocampal astrocytes in slices, as well as cortical astrocytes in vivo. While slice values are similar across regions, in vivo Na⁺ levels are significantly higher. A potential confound is temperature: slice recordings were performed at room temperature, whereas in vivo imaging was at physiological temperature (~37 °C). Since fluorescence lifetime generally decreases with increasing temperature, this factor should be explicitly tested in the calibration.
3. Somatic vs. process comparisons: The authors load astrocytes with dye via a patch pipette and compare somatic and process Na⁺ levels. They conclude that astrocytic processes have higher and more heterogeneous Na⁺ levels. A potential caveat is the reduced photon yield and lower signal-to-noise ratio in fine processes compared to somata, due to smaller volumes and weaker labelling. This should be controlled by calibrating lifetime measurements specifically in processes with known Na⁺ concentrations. Moreover, testing the effect of photon counts (e.g., by summing photons across frames and assessing lifetime variability) would clarify whether instrumental noise contributes significantly to the observed heterogeneity.
4. Relatedly, caution is warranted in interpreting variability within individual astrocytes. Two-photon resolution is limited in XYZ dimensions, and bolus loading may not reliably label fine processes. These limitations might temper the conclusions regarding process heterogeneity.
5. Pharmacological perturbations: The authors examine the effects of various drugs. They find that TTX does not affect Na⁺ levels, whereas gap junction blockade increases Na⁺, and NKA inhibition via low K⁺ dramatically elevates Na⁺. These experiments would benefit from negative controls, such as using a dye or fluorescent protein insensitive to Na⁺. In addition, clarification is needed on why TTX has no effect under basal conditions but does affect Na⁺ when levels are elevated by NKA blockade.
6. Response to K⁺ elevation: The authors report decreased Na⁺ in both somata and processes following high K⁺, with process Na⁺ showing sensitivity to TTX. This suggests compartment-specific regulation by synaptic activity. Since elevating K⁺ engages multiple pathways, complementary experiments using direct synaptic stimulation (electrical or pharmacological) would help isolate and strengthen the claims and mechanisms.
7. Modeling and NKA: The authors use simulations to explore how NKA configuration and expression affect Na⁺. However, the model treats astrocytes as single compartments, while their own data indicate somatic vs. process differences. They acknowledge this, but empirical validation would greatly strengthen the work. For example, testing how knockout/knockdown of specific NKA isoforms affects compartmental Na⁺ levels could directly support the model's predictions and enhance the rigor of their conclusions.

Minor point: Several parameter values are reported in text but not shown as data in the figures. I don't see any reason not to show these data in some format in figures or supplementary figures.

Version 1:

Reviewer comments:

Reviewer #1

(Remarks to the Author)

The authors have improved the manuscript throughout in a way that addresses my concerns and suggestions, and I have no further comments.

(Remarks on code availability)

Reviewer #2

(Remarks to the Author)

The authors have conducted a substantial number of additional experiments to address the concerns raised in the initial review. They have incorporated valuable new data into both the main figures and the supplementary material, and the manuscript text has been carefully revised to reflect these additions. In particular, the inclusion of new experimental data on the differential expression of NKA subunits in the hippocampus, which are integrated into their model, represents a notable improvement.

In my original review, I raised concerns regarding the reliability of the reported in vivo resting astrocytic Na⁺ concentration, which appeared higher than values obtained in acute slices. The authors have now implemented additional controls in their calibration procedure. While it remains challenging to achieve fully reliable calibration under in vivo conditions with current methodologies, the authors provide a clearer description of their approach and more explicitly acknowledge this limitation in the Discussion. It is plausible that basal Na⁺ concentrations are indeed higher in vivo than in slice preparations; however, the precise absolute value is not critical to the main conclusions of the study.

Overall, I thank the authors for their thorough and thoughtful responses to my comments. The manuscript has been significantly improved and is now well suited for publication. I have no further concerns to raise.

(Remarks on code availability)

Reviewer #3

(Remarks to the Author)

The authors have addressed all my concerns. Congrats on a very nice study.

(Remarks on code availability)

Reviewer #4

(Remarks to the Author)

I commend the authors for their excellent work. They have now adequately addressed all of my previous comments. The manuscript is highly compelling and rigorous in its technical and methodological aspects, as well as in the biological phenomena examined and the interpretations presented.

In my opinion, the manuscript is suitable for publication in its present form.

(Remarks on code availability)

Response letter to Meyer et al.: “Cellular and subcellular heterogeneity of astrocytic Na⁺ homeostasis tuning astrocytes into functionally distinct subgroups”

We thank the reviewers for their valuable suggestions and insightful comments, which have helped us to significantly improve the quality of our work. We have highlighted in blue the parts of the manuscript, figures and supplementary information that we have added or substantially modified.

The modifications that we have made include the following:

- additional calibrations of ING-2, adding points in the physiological range of intracellular Na⁺, as well as separate calibrations in astrocyte processes
- an additional elaboration and clarification of the criteria for FL imaging, particularly with regard to smaller cellular compartments
- additional experiments and analysis to consolidate quantitative data on astrocyte Na⁺ as determined by FL imaging with ING-2
- a clearer and more detailed description of *in vivo* imaging and calibration procedures
- additional experiments and analysis to include 'negative controls' in experiments involving different drugs
- further analysis of the relevance of gap junctional coupling on astrocytic Na⁺
- further experiments addressing the relevance of spontaneous neuronal activity on baseline Na⁺
- further experiments and analysis addressing the distribution profile of intracellular Na⁺ in specific conditions
- additional experiments addressing Na⁺ in astrocyte processes
- further experiments addressing the relevance of ionotropic glutamate receptors in astrocyte and neuronal Na⁺ changes induced by inhibition of EAATs as well as in neuronal swelling
- a direct experimental validation of the expression and distribution of different NKA subunits in astrocytes at both the RNA and protein levels, using spatial mRNA detection and immunohistochemistry
- extension of the biophysical model to simulate the spatial profile of Na⁺ in an astrocyte

Overall, we have thus performed a significant number of additional experiments and analyses as well as extended model simulations resulting in 13 new or substantially modified figure panels, one additional full figure, 6 additional supplementary figures and 5 new supplementary tables. Please see our point-by-point response in blue below, which describes the new data included in the revised version of the manuscript in detail. Altogether, the results of our new experiments, analyses and simulations complement and strengthen our previous conclusions.

Reviewer #1:

Meyer et al. present a concise manuscript evaluating subcellular astrocytic Na⁺ concentration *in vitro* and *in vivo*, and explores how intracellular Na⁺ is influenced by a range of factors, most notable glutamate and K⁺ uptake, as well as Na/K-ATPase activity. They also present biophysical modelling results, which strengthen the manuscript, but which lies outside my expertise to evaluate.

This is a highly interesting and potentially very important study that significantly extends our understanding of astrocytic homeostatic functions. I think employing quantitative FLIM analyses as in

the current study is essential to move the astrocyte physiology field forward. While overall I am highly enthusiastic about this study, I have some concerns, in particular about the interpretation of in vivo work and apparent lack of physiological monitoring during in vivo recordings that should – as a minimum – be addressed thoroughly by an elaborate discussion.

Comments and suggestions:

1. According to the In vivo microscopy methods section, only temperature was controlled for during imaging. If more extensive physiological monitoring was performed details must be provided. To be able to interpret physiological parameters from in vivo investigations of anesthetized mice, proper physiological monitoring is crucial, as mouse physiological parameters (especially in the aftermath of acute surgery) rapidly deteriorates. Isoflurane (even though in a relatively low dose was used here) combined with buprenorphine could profoundly affect mouse physiological parameters (as evident when evaluating arterial blood gas composition): Presumably the mice were slightly hypoxic unless oxygen was supplied with isoflurane (not stated), whereas if oxygen was supplied, they are quite likely to be hypercapnic, which again will have consequences for excitability, pH, metabolic state etc. Moreover, within a couple of hours system physiology rapidly deteriorates in mice during general anesthesia (in particular if the mouse has been subjected to acute surgery) and monitoring is of essence to ensure that results could be trusted.

Ideally physiological monitoring should be provided, but as this is primarily a slice study, as a minimum, a thorough discussion of these caveats is warranted.

Our response: We thank the reviewer for raising these important points. In our experiments, isoflurane was delivered together with medical oxygen, and this is now stated explicitly in the revised and significantly expanded method section. In addition to continuous temperature control, we also adjust the isoflurane concentration to keep the animal breathing rate at about 55-65 breaths per minute (Ewald *et al.*, 2011). These details have also been added to the methods section on “In vivo MP-FLIM and analysis”. We fully agree that prolonged anesthesia, particularly following acute surgery, can compromise physiological stability. For this reason, imaging sessions were kept as brief as possible (typically less than 2 hours including the surgery). While we did not perform full arterial blood gas monitoring, we acknowledge the limitations characteristic for anesthetized preparations, including potential alterations in oxygenation, CO₂ levels, pH, and metabolic state. We have added a discussion of these and other specific caveats to the revised manuscript (page 18, lines 10-25).

2. The effects of anesthesia vs the unanesthetized state should be addressed when interpreting the in vivo data vs the in vitro data, and in the discussion. Performing this type of experiment in the unanesthetized state is of course very challenging if not impossible, but even so the interpretation of results must take the direct effects of anesthesia on brain state and the processes studied into account. One would expect that the drugs administered (iso/buprenorphine) would influence both neural activity and neuromodulator activity in dose and drug-dependent ways, in addition to potentially directly affecting ionic pump activity and channels. Hence, the explanation provided of the higher Na⁺ levels in the in vivo situation (“higher levels of neuronal activity”) should be expanded and elaborated to address all these aspects.

Our response: We appreciate the reviewer’s comments regarding the interpretation of our results. We fully agree that buprenorphine and particularly isoflurane can alter several aspects of brain physiology. A discussion of the impact of anesthesia was indeed missing. We have expanded the relevant section in the discussion (page 18, lines 10-25) and added several references that illustrate how specifically isoflurane may contribute to our observations. Unfortunately, there are no

genetically encoded Na⁺ indicators suitable for *in vivo* imaging available to our knowledge, which would enable us to repeat these experiments in awake mice.

3. Slice experiments were performed at room temperature. The effect of temperature on Na⁺ levels (or rather the known effects of temperature on the Na/K-ATPase, GLT1 etc) should be appropriately discussed.

Our response: Thank you for this important suggestion. Our previous research has indeed shown that increasing the temperature by 10°C accelerates the rise time of glutamate-induced Na⁺ transients by a factor of 1.5 in Bergmann glial cells (Bennay *et al.*, 2008). Furthermore, we found that an increase of 10°C reduced the decay constants for recovery from intracellular Na⁺ transients by approximately 1.7-fold (Rose *et al.*, 1999; Bennay *et al.*, 2008; Mondragao *et al.*, 2016). These values are close to the Q10 values reported for the NKA, which supports its expected dominant role in restoring the electrochemical Na⁺ gradient (Skou & Esmann, 1992). As suggested, we have added a brief discussion of this topic to our manuscript (page 21, lines 8-14).

4. The *in vivo* surgical preparation procedure should be present in the methods section, not only references to previous work.

Our response: As requested, we have expanded the *in vivo* part of the methods section (chapter: “*In vivo* MP-FLIM and analysis”) to clarify the surgical preparation procedure for *in vivo* experiments.

5. Slice experiments were performed in mouse pups (likely for good reason to allow easier patch clamp), however this should be discussed when interpreting the results as there are known developmental changes in everything from resting membrane potential, Na/K-ATPase and GLT1 expression etc that could likely influence Na⁺ levels.

Our response: Our physiological experiments were performed in juvenile animals, at postnatal days (P) 14–20. At this time period, several major steps of astrocyte maturation have been completed, and physiological properties (E_m , R_m), and expression of GFAP, NKA, GLT-1 and other proteins are already at, or near, adult levels (Kafitz *et al.*, 2008; Schreiner *et al.*, 2014; Hanson *et al.*, 2015; Larsen *et al.*, 2019). Moreover, using ratiometric widefield Na⁺ imaging, we previously demonstrated that glutamate- and NMDA-induced Na⁺ transients in astrocytes in the hippocampal *stratum radiatum* and cortical layers II/III are indistinguishable between juvenile animals (P14–P20) and adults (P90–P97), indicating that average Na⁺ levels are also similar in these age groups (Ziemens *et al.*, 2019). However, we fully agree that changes in the expression levels of Na⁺-dependent transporters occur until adulthood, which may influence the levels of Na⁺ and Na⁺ signaling in astrocytes and have therefore expanded our discussion as requested (page 18, lines 5-9).

6. As far as I understand, only processes clearly visible by SR101 labelling were analyzed. However, astrocytes have processes that are considerably smaller than these, that cannot be resolved with the optical imaging employed. However, would it not still be possible to assess fluorescence lifetime in regions where these processes reside (“gliopil”) even though they only occupy a fraction of the volume? Or is the sensor not properly filling these smallest of processes?

Our response: Thank you for this comment. Reliable calculation of the fluorescence lifetimes of ING-2 requires a reliable fitting of photon distributions. In our experimental conditions and technical settings, the latter required at least 2000 photons and five photons per pixel per frame, which was usually not obtained in very fine processes or in gliopil using typical regions of interest and typical laser powers to avoid tissue damage. To clarify and illustrate this analysis and criterion, we have

prepared an additional figure (Suppl. Fig. 3), presented in the Supplementary Information, and explain the procedure in more detail in the results (page 7, lines 21-25) and in the methods section (page 26, lines 5-7). Imaging details and analysis criteria were established in initial exploratory experiments for measuring larger populations of astrocytes repeatedly (e.g. for drug application) to capture overall baseline $[Na^+]$ heterogeneity. Careful further adjustments of the imaging parameters, upcoming technical improvements and further refinements of the analysis should allow us, in the future, to obtain detailed information also at the level of the finer astrocyte processes.

7. Even with the elaborate supplementary statistical summary it is somewhat difficult to understand which n/N has been used in the various comparisons. An example to illustrate: in Figure 6m, were recordings of cells directly compared (i.e. n = number of cells) or was the data binned (per slice/mouse or similar)? A potentially more sound approach that strikes a better balance between being too conservative (comparing means per slice) vs too likely to detect significant differences (n = observations) would be to employ a method that takes into account the hierarchical structure of the data where there are many observations per slice, and several slices per mouse (e.g. mixed effects linear regression models).

Our response: Thank you for pointing this out. Throughout the manuscript, 'n' now refers to the number of analyzed cells, 'N' to the number of slice preparations *in situ*, and 'n_p' to the number of individual processes analyzed. This statement has been added to the Methods section (page 23, lines 17/18), and the text has been clarified throughout.

As suggested, we also performed a mixed linear regression analysis to test for the potential confounding effects and interdependencies of the Na^+ levels obtained from individual cells on the respective slice preparation and/or animal. This analysis revealed that there is no significant relationship between $[Na^+]$, of astrocytes and the number of slice preparations or mice used ($p=0.586$). We have added this additional test procedure and information to the methods and results of the manuscript (page 22/23, lines 25-2; page 5, lines 19-20).

Reviewer #2

The manuscript by Meyer et al describes the use of 2-photon fluorescence lifetime imaging for assessing astrocytic Na^+ in acute brain slices and *in vivo*. To perform their study, the authors used the commercially available indicator ING-2. Using a pulsed infrared Ti:Sapphire laser coupled to time-resolved detectors, they detected and calibrated intracellular Na^+ concentrations in astrocytes *in situ*. Of note, the same group previously published the detailed use of this approach in organotypic slice cultures (Meyer et al 2022) applied to neuronal Na^+ assessment.

The presented approach convincingly demonstrates its ability to accurately monitoring astrocytic Na^+ concentrations, and classical manipulations of intracellular Na^+ (e.g. using low or high bath K^+) led to expected changes. Using their approach, the authors found that astrocytes do not present a uniform Na^+ , contrary to previous results obtained from cultured astrocytes, notably by the same group. A bimodal distribution of resting Na^+ was found in the population of astrocytes, interpreted by the authors as being likely linked to their differential expression of sodium pump isoforms (i.e. $\alpha 2\beta 1$ vs $\alpha 2\beta 2$), as supported by their mathematical modeling. They also found Na^+ to be higher in astrocyte processes compared to somata, and that manipulations of sodium pump activity by external K^+ has more pronounced effects in processes.

The study provides valuable insight into astrocytic Na⁺ in situ and in vivo. It must be noted that very few attempts of quantitative assessments of ion concentrations in cells within the tissue using imaging approaches have been reliably developed. The proposed approach using FLIM, rather than intensity-based signal detection, appears to robustly achieve this goal.

Arguably, the study seems to provide mainly incremental knowledge to the field and to a some extent confirms reasonable expectations. The potential role of different subunit compositions of the sodium pump in adjusting resting Na⁺ levels differentially in individual cells is sound and supported by the presented simulations. However, the lack of direct experimental validation limits the strength of this conclusion and leaves the central hypothesis insufficiently substantiated.

The paper is well written and interesting to read. This technically challenging study was carefully performed and provides a detailed report on use of FLIM for assessing intracellular Na⁺ of astrocytes in situ or in vivo.

Our response: Thank you for your positive overall evaluation of our work. We would like to point out that the heterogeneity in intra- and intercellular [Na⁺] of astrocytes reported by our study had not been experimentally described before and is, in fact, rather unexpected. Na⁺ is a highly mobile, unbuffered ion that has been shown to move rapidly between individual, gap-junction coupled astrocytes in culture and in brain slices (e. g., Rose & Ransom, 1997; Langer *et al.*, 2012; Langer *et al.*, 2017). This is in stark contrast to intracellular Ca²⁺, for example, for which concentration differences between individual cells and cellular microdomains have been demonstrated experimentally in many cell types, including astrocytes (e. g., Volterra *et al.*, 2014; Shigetomi *et al.*, 2016; Bindocci *et al.*, 2017; King *et al.*, 2020; Semyanov *et al.*, 2020). Given the high intracellular Na⁺ diffusion coefficient ($D_{Na^+} = 600 \mu\text{m}^2/\text{s}$; Nelson *et al.*, 2025), such cellular and subcellular heterogeneity was not expected (nor experimentally demonstrated) before for baseline astrocyte [Na⁺].

In addition, based on the reviewers' comments and suggestions, we have extensively revised our manuscript, adding a significant number of new experiments and analyses as well as extended model simulations. Importantly, this includes a direct experimental validation of the expression and distribution of different NKA subunits in astrocytes at both the RNA and protein levels, using spatial mRNA detection and immunohistochemistry, which is detailed in our response to comment 3. We would also like to refer the reviewer to the changes and additions we made in response to comments and suggestions from other reviewers. Altogether, the results of our new experiments, analyses and simulations complement and strengthen our previous conclusions.

Comments:

1) - Local syncytium: to what extent the locally connected astrocytes by gap-junction would actually share common Na⁺ levels? In other words, it is not clear if astrocytes with different Na⁺ levels observed in the study are not those more distant to each other or belonging to different syncytia. Previous studies have shown astrocytes to be extensively connected by gap junctions, a cellular organization for instance central to the regulation of interstitial K⁺ (potassium buffering). Is Na⁺ differently able to diffuse from cell to cell than other ions or small dye molecules?

Our response: The extent to which locally connected astrocytes would share similar Na⁺ levels is a very interesting question. As mentioned above, Na⁺ is a highly mobile, unbuffered ion that has been shown to move rapidly between individual gap-junction-coupled astrocytes in culture and in brain slices (e. g., Rose & Ransom, 1997; Langer *et al.*, 2012; Langer *et al.*, 2017), which is why it was generally assumed that Na⁺ was equalized between gap-junction-coupled cells.

To further address this question, we followed the reviewer's suggestion and analyzed the difference in $[Na^+]_i$ in neighboring astrocytes in relation to their distance from each other within a given slice preparation. We found that the variance of $[Na^+]_i$ was smallest for cells within 50 μm of each other, increasing significantly with greater distances. This indicates that the $[Na^+]_i$ of an astrocyte is at least partly dependent on its spatial position or whether it belongs to a special gap junction-coupled syncytium. This new information has been added to the results (page 10, lines 4-10 and new Figure panels 4g-i).

2)- Local neuronal activity: individual astrocytes are embedded into local microcircuits, i.e. possibly surrounded by varying levels of electrical activity of neurons differentially releasing K^+ or neurotransmitters. Could this explain different Na^+ levels found in single astrocytes? It is for instance not clear whether the heterogeneity or bimodal distribution of Na^+ levels persists in the when silencing neuronal activity (TTX).

Our response: We have performed additional experiments, analyses and mathematical simulations to address these valid points. Our new experimental results show that perfusion of brain tissue slices with TTX did not result in detectable differences in the baseline $[Na^+]_i$ of somata nor of processes of astrocytes. In addition, the bimodal distribution of $[Na^+]_i$ remained. These results, now illustrated in new Figs. 4a and 4b, suggest that the different levels of baseline $[Na^+]_i$ in astrocytes are not primarily due to local spontaneous activity of microcircuits in brain tissue slices resulting in the local release of K^+ and glutamate. It is worth noting that the release of glutamate activates Na^+ import by EAATs, while the release of K^+ activates NKA and Na^+ export, thereby counteracting the influx of Na^+ related to EAATs. It is therefore the sum of these opposing processes that determines the concentration of Na^+ in astrocytes. These new points have been added to the results (page 9, lines 11-20).

3)- Differential expression of sodium pump isoforms: biophysical modelling presented and discussed by the authors supports a direct effect of isoform composition on Na^+ levels of individual astrocytes. To strengthen this hypothesis, the authors should provide direct assessment at the level of individual cells, of the existence of differential subunit composition, backed up e.g. by immuno stainings or spatial omics approaches, and ideally connect these results to the observed Na^+ level in those cells. Such experiments could also address the question of whether the pump isoform composition differs between somata and processes, which could contribute to the observed different Na^+ levels between the two compartments.

Our response: Thank you for this comment. As suggested, we have performed additional experiments directly addressing the differential expression of Na^+ pump isoforms in astrocytes. We fully agree that, ideally, the information on the Na^+ concentration in the soma and processes of a given astrocyte should be combined with experiments that address the spatial pattern of NKA isoform composition and distribution within the same cell. However, determining the concentration of Na^+ in the somata and processes of individual astrocytes using FLIM requires cells in thick (250 μm) acute tissue slices to be loaded with two fluorescent dyes (SR101 and ING-2). In addition, a third marker is needed for a later unbiased identification of selected astrocytes. After live imaging of Na^+ , these slices then need to be fixed and re-sliced to allow penetration of labelling compounds for RNAScope or IHC with three to four additional markers. Subsequently, a very sophisticated 3D reconstruction of the re-sliced serial sections is required to address all cells including their processes that were studied with FLIM in the thick slice. In a considerable number of cases, such a reconstruction will fail because of loss of tissue and clear staining patterns by re-sectioning.

This complex and highly demanding protocol therefore results in a low yield of reliably marked and evaluable tissue, greatly reducing the success rate of RNAScope or IHC staining in the astrocytes and processes in which a successful FLIM-based determination of Na⁺ had previously been performed. This is particularly true for membrane transport proteins, such as the specific NKA subunits of interest here. We therefore decided against such a combined approach in favor of optimal staining results. As described below, we obtained meaningful and high-quality staining patterns at both the RNA and protein levels in freshly cut and immediately fixed tissue. In the future, we will aim to adapt cellular live imaging and protocols for (sub-)cellular staining in a way, that a combination of both is possible. However, we expect that achieving this successfully will require a substantial amount of time and effort.

To address the differential expression of Na⁺ pump isoforms in astrocytes in our present study, we probed hippocampal sections for mRNA using RNAScope (Karpf *et al.*, 2022) and for protein expression using immunohistochemistry (IHC). In summary, these demonstrate the presence of NKA β 1 and β 2 subunits at the mRNA and protein level in GFAP-positive cells, confirming their expression in astrocytes. In general, the mRNA and protein staining patterns align with the reported lower expression levels of β 1 compared to β 2. Furthermore, the staining patterns are compatible with the view that processes exhibit higher expression of β 2 than of β 1. These new results are now described on pages 13-14 and illustrated in the new Fig. 7 as well as the new Suppl. Fig. 8.

In order to further address the impact of variations in NKA isoform composition between somata and processes on intracellular Na⁺ levels, we also extended the biophysical model. We incorporated a reconstructed morphology of an astrocyte (obtained from Neuromorpho.org; RRID:SCR_002145; Kodali *et al.*, 2021; morphology #NMO_282188), and modelled the spatial profiles of Na⁺ (and other ions) in the cell. We varied the relative expression of NKA α 2 β 1 and α 2 β 2 isoforms according to distance from the soma. These new simulations show that the spatial heterogeneity of [Na⁺] between the soma and processes of individual astrocytes can be replicated by modelling NKA expression and isoform composition, respectively as a function of distance from the soma. The additions made to the model, along with the results of the new simulations, have been incorporated into the revised manuscript (page 16, lines 12-24; new Fig. 8d; Suppl. Methods Text 2).

Altogether, RNAScope and immunohistochemistry thus indicated differential spatial expression patterns of NKA β 1 and β 2 subunits in astrocytes. Moreover, the additional biophysical modelling of spatially differential NKA expression together with varying strength of Na⁺ influx replicated the experimentally observed heterogeneity in astrocytic [Na⁺]. These new results thus strengthen our conclusions that the differential NKA isoform composition locally adapts astrocytic Na⁺ homeostasis to the specific requirements of surrounding neural networks.

4)- p. 6, line 1: Effects of lowering external Na⁺: It would be worth mentioning how long it takes for cells to reach this lowest level after the switch to zero external Na⁺.

Our response: It took about 2 minutes before FL in astrocyte somata had reached values near 0 mM [Na⁺] in Na⁺-free calibration saline. We have added this information to the text (page 6, line 5).

5)- p. 6, line 9: The study in vivo was performed with anesthetized head-fixed mice. Anesthesia is known to have a strong effects on astrocyte Ca²⁺ signaling compared to awake head-fixed mice. Do the authors have information whether this also applies to Na⁺ signaling or can the authors comment on whether such effect is expected as well for Na⁺?

Our response: That is a great question. Indeed, isoflurane anesthesia is known to dampen astrocytic Ca²⁺ transients in astrocytes (Thrane *et al.*, 2012), which matches our observations in other,

unrelated experiments. However, it has not been studied systematically to our knowledge if and how the astrocytic resting $[Ca^{2+}]$ correlates with the depth of anesthesia. It is therefore difficult to compare our analysis of resting $[Na^+]$ with these observations. In preliminary experiments, we in fact attempted to detect spontaneous fluctuations of $[Na^+]$ *in vivo* which are expected because of activation of Na^+ -dependent glutamate uptake after its synaptic release for instance. However, we could not resolve these at the soma because the Na^+ -transients are small relative to the baseline $[Na^+]$ (which is very different for Ca^{2+}) and/or because they are distributed in time and space throughout the arborization of the astrocytes during synaptic activity *in vivo* and thereby escape detection (e.g., (Langer & Rose, 2009; Ziemens *et al.*, 2019). With the help of genetically-encoded Na^+ indicators, such questions could be addressed in awake mice. However, such indicators do not exist currently as far as we are aware. We have added some of these thoughts to the discussion (page 18, lines 10-25).

6)- p. 9, starting last paragraph: Glutamate transporter inhibition and baseline Na^+ levels:

- The reviewer fails to see what the experiments using TBOA bring to the question of baseline Na^+ values of astrocytes. It is for instance not described whether TBOA application influences the heterogeneity and bimodal distribution of intracellular Na^+ in astrocytes.

Our response: Thank you for this remark. The experiments using (TFB-)TBOA (please see below) were performed to analyze the role of Na^+ -dependent glutamate transporters (EAATs) in setting baseline $[Na^+]$ in astrocytes. Indeed, our data suggest that baseline $[Na^+]$ is at least partly determined by Na^+ influx through EAATs.

As suggested, we carried out additional experiments and analyses to address the reviewer's question regarding the distribution of Na^+ levels in the presence of (TFB-)TBOA. Interestingly, these show that $[Na^+]$ shifted towards a unimodal distribution following the application of (TFB-)TBOA (new Fig. 5d). This shift in distribution and the convergence of the two former populations is most likely to be related to the fact that the decrease in Na^+ induced by (TFB-)TBOA depended on the former baseline $[Na^+]$, and NKA expression, respectively (please see Fig. 5c). To test the plausibility of this finding, we also performed additional simulations, in which we changed the subunit composition and/or expression of NKA together with the strength of Na^+ influx (new Suppl. Table 5). These show that the effect on $[Na^+]$ was strongest in cells predominantly expressing $\alpha 2\beta 2$. For example, decreasing Na^+ influx to 60% decreased $[Na^+]$ of astrocytes with 100% $\alpha 2\beta 2$ expression to levels close to those of astrocytes that exclusively express $\alpha 2\beta 1$ (new Suppl. Table 5). This led to a shift to the left and a narrowing of the $[Na^+]$ distribution, comparable to that observed experimentally when EAATs were inhibited by TFB-TBOA (new Fig. 5d). These new results and statements have been added to the text (page 11, lines 21-23; new Fig. 5d; page 15, lines 11-17; new Suppl. Table 5).

- Swelling of neuronal cell bodies after TBOA application: this effects most likely relates to glutamate excitotoxicity, as probably also assumed by the authors. Can the huge Na^+ increase found in neurons (Fig 5) be prevented by blockade of glutamate receptor blocker? How to interpret those experiments in the context of the initial question of the determinants of baseline Na^+ in astrocytes?

Our response: We fully agree with the reviewer's view that TFB-TBOA induces glutamate excitotoxicity because inhibiting glutamate transport immediately increases extracellular glutamate (Rothstein *et al.*, 1996; Jaubaudon *et al.*, 1999).

As suggested, we performed new experiments to address the role of glutamate receptors in neuronal Na^+ elevation. We found that in the presence of CNQX and APV, which block ionotropic glutamate receptors, the (TFB-)TBOA-induced increase in neuronal $[Na^+]$ was significantly reduced

(illustrated in new Figs. 5e, f). Moreover, we analyzed the swelling of neuronal cell bodies, which was significantly reduced in the presence of the receptor blockers as well (illustrated in new Suppl. Fig. 7). These experiments emphasize the vital role of ionotropic glutamate receptors in neuronal cell swelling and Na^+ loading in the presence of EAAT blockers. They also demonstrate that, during the initial stages of extracellular glutamate accumulation, severe neuronal Na^+ loading and excitotoxic neuronal swelling, astrocytes can maintain low levels of intracellular $[\text{Na}^+]$. These new results and statements have been added to the text (page 11-12, lines 24-6).

- TBOA is a valuable non-transported competitive inhibitor of glutamate transporters. However, its affinity for EAATs is relatively low, compared to e.g. TFB-TBOA. The authors used 1micromolar TBOA, a concentration much lower than the published EC50 values, and to the one the same authors used previously (200micromolar). The choice of this low concentration should be discussed.

Our response: Thank you for pointing this out. We would like to clarify that we used TFB-TBOA, however, we did not indicate this correctly. We apologize for this error and have corrected it in the revised version of our manuscript.

7) - In vivo astrocytic cytosolic Na^+ : the authors write that calibration of ING-2 lifetimes for in vivo experiments was performed in a cuvette using calibration solutions without ionophores. While they claim that minor differences between fluorescence lifetime components observed in the cuvette and in vivo (p.22, lines 12-15), in their previous study using FLIM for neuronal Na^+ (Meyer, J Neurosci 2022), they reported an apparent K_D of 8.9mM in cuvettes, while in situ calibration in the neuronal cytosol yielded an apparent K_D of >33mM, concluding that that the apparent K_D of chemical sodium indicators is significantly higher inside cells as compared with cuvette calibration condition. If the same applied to the astrocyte Na^+ of the present study, it could mean that the apparent higher Na^+ levels claimed to be present in the in vivo brain compared to slices is due to this difference in K_D . The authors should address this point and the validity of the absolute in vivo Na^+ concentrations values presented.

Our response: Thank you for pointing out this potential issue. The *in vivo* and *in situ* data sets were recorded at different setups and in different labs and while theoretically dye properties should be independent of the equipment used, considerable lab-to-lab and setup-to-setup variability of calibration values is common. For example, for the imaging rig used for slice experiments in the present study, we obtained an apparent *in situ* K_D of 21 mM. To consolidate this value further for the present *in situ* dataset, we performed additional *in situ* calibration experiments, adding calibration points at 5, 10, 15, and 25 mM Na^+ , i.e., in the lower concentration range. The revised Fig. 1e illustrates that the new calibrations confirmed this rig-specific apparent K_D for somata and astrocyte processes, which was used to convert all FL data obtained in the slice experiments into $[\text{Na}^+]$.

Unfortunately, it is not possible to perform reliable intracellular calibration *in vivo* in the intact brain. Simply applying the *in situ* calibration parameters (K_{Dapp} : 21 mM) obtained at a different rig to the data acquired on the *in vivo* system produced nonsensical values (below 0 mM) in 40% of astrocytes, which is perhaps not surprising. Therefore, we used the calibration in a cuvette from the *in vivo* microscope for quantifying the somatic $[\text{Na}^+]$ *in vivo*. To support the validity of this approach, we also tested for differences in the decay times of the individual lifetime components between both conditions (in a cuvette on the *in vivo* setup vs. *in vivo*), because lifetime components can be significantly affected by the environment of the dye. Indeed, we found small but detectable differences. We then determined the quantitative effect of these differences on our $[\text{Na}^+]$ estimates using an approach previously employed for Ca^{2+} FLIM (King *et al.*, 2020) and found that the effect on

the calculated $[Na^+]$ is negligible. To clarify this, we have added these points and now explain this procedure in detail in the expanded Methods section on *in vivo* imaging. We would also like to point out that significant differences in dye properties may exist between slice preparations and *in vivo* due to variations in intracellular pO_2 , pCO_2 , metabolic state, and so on.

Reviewer #3

This manuscript uses exciting new imaging technologies to quantify astrocyte intracellular Na in situ and in vivo. This is very important work that provides information lacking in the field, as very little is known about the intracellular concentration of many ions in astrocytes. The work is highly rigorous, technically sound, and innovative. I am very enthusiastic about this study and am confident it will have a strong impact on the field. There are a number of moderate/minor concerns, in particular with how Na concentration was calculated and validated, that would greatly increase the acceptance of the quantifications as valid and accurate.

1)- The results say that the in vitro calibration curve was adjusted for 2P in vivo and is described in the methods. I did not see that section in the methods. Please add/update to include a description of how in vivo $[Na^+]$ was calculated.

Our response: We apologize for the oversight. Important details regarding the calculation of $[Na^+]$ *in vivo* were indeed missing. We have revised the methods section, which now includes a detailed description of how *in vivo* data was analyzed.

2)- The most exciting aspect of this study is the intracellular quantification of astrocyte intracellular $[Na^+]$, so ensuring this calibration and calculation is correct is critical. The study would be strengthened by using neuron AP peaks, which can be calculated based on equilibrium potentials, as an independent validation that the $[Na^+]$ calculations in astrocytes are correct.

Our response: Thank you for this suggestion. As requested, we have performed further experiments to strengthen the validity of our calibration and calculations. However, we opted for alternative approaches to the one suggested by the reviewer for several reasons: APs do not reach E_{Na^+} , and evaluating Na^+ changes during a single AP is technically not feasible because the spatiotemporal resolution of Na^+ imaging is too low, particularly when FLIM is used (Rose *et al.*, 1999; Fleidervish *et al.*, 2010; Baranauskas *et al.*, 2013; Kotler *et al.*, 2023). Moreover, repetitive AP firing only increases somatic $[Na^+]$ by a few mM, as Na^+ rapidly dilutes within neuronal cell bodies and Na^+ export by the NKA masks its influx (Rose *et al.*, 1999).

Instead, we first depleted the tissue slices of Na^+ by perfusing them with Na^+ -free saline. This caused a slow decline in fluorescence lifetimes (FL) to a new stable level, indicating a nominal depletion of cells from Na^+ . We then switched to Na^+ -free calibration saline and determined the FL again. We found that the cellular FL in cells that were depleted of Na^+ corresponded to that in calibration saline devoid of Na^+ , indicating that the FL for 0 mM Na^+ obtained in calibration experiments is correct. This new data is described in the text (page 6, lines 5-9) and shown in the new Fig. 1j.

For a second independent validation, we performed experiments and calibrations using intensity-based, widefield imaging with the ratiometric Na^+ indicator SBFI (sodium-binding benzofuran isophthalate) in SR101-stained astrocytes in tissue slices. With this Na^+ -indicator and

imaging modality, we determined an average baseline $[Na^+]$ of 13.1 mM in astrocyte somata, which is similar to that obtained by FL imaging of ING-2 and also to values reported earlier from different laboratories (e. g., Chatton *et al.*, 2016). Notably, the distribution of astrocytic Na^+ as determined by ratiometric intensity-based imaging with SBFI was best fit by a bimodal distribution, again confirming the results obtained by FLIM of ING-2. These results not only verify those obtained by quantitative FLIM of ING-2. In addition, they for the first time show that the absolute fluorescence ratio of SBFI, as determined by widefield imaging, can be used to report baseline $[Na^+]$. These new results have been added to the text (page 6, lines 9-15) and are illustrated in the new Suppl. Fig. 1.

Lastly, also we performed additional calibrations of ING-2 FL, adding calibration points at 5, 10, 15, and 25 mM to consolidate the apparent *in situ* K_D of 21 mM. The revised Fig. 1e illustrates calibrations confirmed this rig-specific apparent K_D for somata and astrocyte processes, which was used to convert all FL data obtained in the slice experiments into $[Na^+]$.

3) - Does ING-2 load astrocytes *in vivo*? In the 2P image shown, ING-2 looks very diffuse. Where is the dye located in this prep? The study would benefit from a reconstruction of an SR101 or genetically labeled astrocyte *in vivo* demonstrating ING-2 loading.

Our response: We thank the reviewer for this suggestion. We now include a zoomed-in supplementary two-photon excitation fluorescence image (see new Suppl. Fig. 2) showing that ING-2 AM loads both neurons and astrocytes *in vivo*. Because of the diffuse labelling of the neuropil, we can reliably analyze its fluorescence only in the cell bodies of astrocytes and neurons but not in their processes *in vivo*. Quantification of fluorescence lifetimes *in vivo* was performed for somatic ROIs and converted to somatic Na^+ concentrations, indicated as $[Na^+]_s$. We have added this and further details to the methods section (chapter "In vivo MP-FLIM and analysis").

4)- If the intracellular solution dialyzes Na in the soma, shouldn't it do the same in the processes? Pipette withdrawal addresses this nicely, but is still difficult to understand. The paper cites a study showing similar findings in neurons, but references a pre-print from the Rose lab. Is there other data supporting this approach?

Our response: To the best of our knowledge, this has not been demonstrated before. Compared to cellular Ca^{2+} regulation and signaling in the nervous system, intracellular Na^+ homeostasis is largely understudied. As far as we are aware, the mentioned study was also the first to perform FLIM of intracellular $[Na^+]$ in neuronal dendrites in tissue slices. This manuscript has now been published in the *Journal of Neuroscience* (Nelson *et al.*, 2025) and the reference has been updated accordingly. Interestingly, in our former study performed in neurons, we found no difference in baseline $[Na^+]$ between the soma and dendrites, nor between dendrites at different distances from the soma nor between different dendrite orders (Nelson *et al.*, 2025). In contrast, our current FLIM study of astrocytes revealed an increased baseline $[Na^+]$ in astrocyte processes. Our data and simulations suggest that this can be explained by either an increased influx of Na^+ or by different expression of NKA isoforms in the distal processes (see new Figs. 7 and 8d). Our results therefore suggest that astrocyte processes regulate $[Na^+]$ independently of the soma, a finding similar to that described in previous FLIM studies on astrocyte Ca^{2+} (Zheng *et al.*, 2015) or Cl^- (Weilinger *et al.*, 2022). To clarify this, we expanded the discussion on this topic (e.g., page 20, lines 9-11).

5) - The change in extracellular K in response to carbanoxelone is so tiny that its relevance is questionable. On the other hand, the authors report that TBOA increases K(e) by 2.5 mM, a highly

relevant increase. Is this a statistically significant increase? Reporting these 2 changes should be contextualized better.

Our response: We apologize for not stating the results of our statistical tests on the (TFB-)TBOA-induced increase in extracellular K^+ in the text, which was highly significant. This information is now included in the manuscript and statistical summary file.

Moreover, we have prepared an additional supplementary figure (new Suppl. Fig. 6) to illustrate data not shown in our original manuscript, including bar charts summarizing the CBX-induced increase in extracellular K^+ .

6) - How does low K^+ drive neuronal activity to make increases in Na^+ affected by TTX? I would expect low K^+ to decrease neuronal activity.

Our response: Thank you for this comment. When revisiting the results of this set of experiments, we noticed that the p -value in the previous Fig. 4i had been incorrectly indicated as “**” (two stars) when it should have been “*” (one star). We apologize for this error in the illustration of the data.

To address the reviewer’s question, we have performed additional electrophysiological recordings on CA1 pyramidal neurons. These confirmed the expected neuronal hyperpolarization upon perfusion with low K^+ . However, upon wash-in of standard extracellular K^+ , neurons rapidly repolarized and then underwent a prolonged phase of increased activity, which partly overlapped with the time window in which most astrocytes showed their peak increase in $[Na^+]$. TTX prevented the phase of increased neuronal activity and action potential firing upon restoring standard extracellular K^+ . This new data is now described in the results (page 10, lines 13-16 and page 11, lines 1-4) and shown in the inset of the new Fig. 4k.

7) - What is the mechanisms by which TBOA causes such significant increases in neuron Na^+ and swelling? Increased neuronal activity? Is this effect TTX dependent also?

Our response: Inhibition of glutamate transport causes an immediate increase in extracellular glutamate (Rothstein *et al.*, 1996; Jambaudon *et al.*, 1999). In response to the reviewer’s suggestion, we performed additional experiments in the presence of CNQX and APV, which block ionotropic glutamate receptors. We found that the increase in neuronal $[Na^+]$ induced by (TFB-)TBOA was significantly reduced under these conditions (illustrated in new Figs. 5e, f). Moreover, we analyzed the swelling of neuronal cell bodies, which was significantly reduced in the presence of the receptor blockers as well (illustrated in new Suppl. Fig. 7). These experiments emphasize the vital role of ionotropic glutamate receptors in neuronal cell swelling and Na^+ loading in the presence of EAAT blockers. They also demonstrate that, during the initial stages of extracellular glutamate accumulation, severe neuronal Na^+ loading and excitotoxic neuronal swelling, astrocytes can maintain low levels of intracellular $[Na^+]$. These new results and statements have been added to the text (page 11-12, lines 24-6).

8) - How do other manipulations tested (low K^+ , carbenoxeline, TBOA) affect Na^+ in the processes of astrocytes? Many manipulations only report Na^+ changes in the soma.

Our response: In response to the reviewer’s question, we have performed further experiments to determine the effect of (TFB-)TBOA in astrocytes processes. As described above in response to comment #7, inhibition of EAATs by (TFB-)TBOA was accompanied by significant neuronal swelling. Unfortunately, the resulting tissue movement caused astrocytic processes to move within or out of the focal plane. This resulted in a loss of fluorescence and reduction in photon counts, often below the critical value necessary to allow reliable fitting of photon distributions (see Methods). In regions

of interest covering astrocyte processes that remained above this value, the reduction in photon counts caused a decrease in the signal-to-noise ratio from about 2-3 mM to below 10 mM Na⁺. This is why we abandoned these measurements.

In another set of experiments, we probed for the effect of TTX on [Na⁺] in processes. Our new experimental results show that perfusion of brain tissue slices with TTX did not result in detectable differences in the baseline [Na⁺] of processes of astrocytes. These results, now illustrated in the new Figs. 4a, suggest that the different levels of baseline [Na⁺] in astrocyte processes are not primarily due to the spontaneous activity of microcircuits in brain tissue slices at rest resulting in the local release of K⁺ and glutamate. It is worth noting that the release of glutamate activates Na⁺ import by EAATs, while the release of K⁺ activates NKA and Na⁺ export, thereby counteracting the influx of Na⁺ related to EAATs. It is therefore the sum of these opposing processes that determines the concentration of Na⁺ in astrocytes. These new points have been added to the results (page 9, lines 13-20).

- Do the authors postulate that intracellular Na⁺ is the limiting factor in ATPase activity – when Na⁺ goes below a threshold does ATPase activity decrease/stop?

Our response: Binding of three intracellular Na⁺ is a critical step of the Post-Albers model of the pump cycle (Pietrobon & Conti, 2024). Our interactive 3D-Plots (Link) show that reducing [Na⁺]_i to 2 mM decreases its activity to 2% of V_{max} in α2β1 and to 10% of V_{max} in α2β2 expressing astrocytes, respectively, illustrating this Na⁺-dependence. To clarify this, we have revised Suppl. Fig. 9 (former Suppl. Fig. 2) and now describe these exemplary numbers in the results (page 16, lines 9-11).

Minor concerns:

- How does perfusion of Na⁺ solutions onto slices lead to equilibrated intracellular Na⁺? The methods describe the use of gramicidin but this should be noted in the results section, at least briefly.

Our response: Gramicidin is a sodium ionophore, i.e. a channel-forming antibiotic peptide. As requested, we have added this information to the methods (page 25, lines 7-10) and results section (page 5, lines 8-11).

- The paper uses an AM version of ING-2, but this is only mentioned in the methods. Again, this should be mentioned in the results at least briefly.

Our response: We have added this information as requested (page 5, line 5).

Reviewer #4

The manuscript describes the use of fluorescence lifetime-based imaging of astrocytes to determine the basal levels, compartmentalization, and regulation of Na⁺ levels, using both brain slice preparations and in vivo imaging in the mouse cortex. The authors use the unique ability of lifetime imaging to quantify and compare absolute Na⁺ levels, their response to pharmacological perturbations, and their cellular and subcellular heterogeneity. Their findings suggest the existence of significant variation in Na⁺ levels—particularly within astrocytic processes and branches—which has not been well characterized previously. The presence of small Na⁺ signaling domains in astrocytic processes may support critical local mechanisms, such as the modulation of glutamate spillover and synaptic remodeling.

Currently, there are technical limitations in addressing these questions. No genetically encoded Na⁺ indicators with sufficient signal-to-noise ratio are available to monitor Na⁺ levels. Furthermore, as the authors point out, intensity-dependent dyes present major challenges for quantifying absolute analyte concentrations, especially in the fine microdomains of astrocytic processes. To overcome this, the authors employ a Na⁺-sensitive dye and introduce an experimental approach to accurately quantify physiological Na⁺ levels in astrocytes. Two-photon fluorescence lifetime imaging microscopy (2pFLIM) is a particularly powerful and appropriate technique for this application. Overall, the authors present valuable information in a clear and concise manner. I have a few comments and suggestions that could improve the rigor and impact of the study:

1. Calibration (Fig. 1): The authors show a dose-response curve to calibrate fluorescence lifetime against known Na⁺ concentrations, which forms the basis for their quantification of basal Na⁺ levels in astrocyte somata. They report heterogeneity among cells, with median Na⁺ levels falling toward the lower end of their calibration curve. Although the physiological values appear to fall within the linear range of the dye, calibration is based on only 1–2 data points in this critical range. Given the ~0.2 ns lifetime change observed here, more careful calibration is warranted to ensure accurate conversion from lifetime to Na⁺ concentration.

Our response: Thank you for this comment. As suggested, we have performed additional *in situ* calibration experiments, adding points at 5, 10, 15, and 25 mM Na⁺. Including these new data points had virtually no effect on the best fit compared to the former calibration curve. The revised Fig. 1e illustrates that the new calibrations therefore confirmed the former rig-specific apparent K_D of 21 mM for somata and astrocyte processes, which was used to convert all FL data obtained in the slice experiments into [Na⁺].

2. Slice vs. *in vivo* comparisons: The authors compare Na⁺ levels in cortical and hippocampal astrocytes in slices, as well as cortical astrocytes *in vivo*. While slice values are similar across regions, *in vivo* Na⁺ levels are significantly higher. A potential confound is temperature: slice recordings were performed at room temperature, whereas *in vivo* imaging was at physiological temperature (~37 °C). Since fluorescence lifetime generally decreases with increasing temperature, this factor should be explicitly tested in the calibration.

Our response: Thank you for this important remark. We have addressed the influence of temperature on ING-2 fluorescence lifetime in our former work on neurons (Meyer *et al.*, 2022). In the latter study, we demonstrated that the relationship between ING-2 FL (τ_{AVG}) and intracellular [Na⁺] is comparable at room temperature and near-physiological temperature (33 °C). This strongly suggests that the observed difference in astrocytic [Na⁺] between slice preparations and *in vivo* is not primarily caused by the temperature-sensitivity of the dye. We have added this information to the discussion (page 18, lines 11-13).

3. Somatic vs. process comparisons: The authors load astrocytes with dye via a patch pipette and compare somatic and process Na⁺ levels. They conclude that astrocytic processes have higher and more heterogeneous Na⁺ levels. A potential caveat is the reduced photon yield and lower signal-to-noise ratio in fine processes compared to somata, due to smaller volumes and weaker labelling. This should be controlled by calibrating lifetime measurements specifically in processes with known Na⁺ concentrations. Moreover, testing the effect of photon counts (e.g., by summing photons across

frames and assessing lifetime variability) would clarify whether instrumental noise contributes significantly to the observed heterogeneity.

Our response: We fully agree that reliable calculation of the fluorescence lifetimes of ING-2 requires a reliable fitting of photon distributions. In our experimental conditions and technical setting, the latter required at least five photons per pixel and frame, and a minimum number of dye molecules in a given structure, respectively. Only processes that met this criterium were included in the analysis. This generally included primary and some secondary processes close to the soma, but not very fine/distal processes. To clarify and illustrate this analysis and criterion, we have prepared an additional figure (Suppl. Fig. 3), presented in the Supplementary Information, and explain the procedure in more details in the results section (page 7, lines 21-25) and in the methods section (page 26, lines 5-7).

As suggested, we also analyzed changes in FL specifically in astrocyte processes in a new set of *in situ* calibration experiments covering $[Na^+]$ between 0 and 30 mM. As illustrated in the revised Fig. 1e, there was no meaningful difference in the best fits covering the data points obtained from the processes compared to astrocyte somata, nor was there a significant difference compared to the fit of the full calibrations performed in neuronal and astrocyte somata.

Moreover, as suggested, we tested the effect of photon counts on the variability of FL. The new Suppl. Fig. 3 shows that summing up photons across frames by increasing the binning from 1 to 10, resulted in nearly identical mean lifetimes, demonstrating that our minimum criterium for FL fitting was sufficient for the reliable fitting of photon distributions.

Finally, we compared $[Na^+]$ calculated from ING-2 FL in processes loaded via whole-cell patch-clamp with those from processes stained by bolus-loading and found no significant difference between the two groups (compare Figs. 3b and 3e). This again demonstrates that the reduced dye concentration and photon yield in bolus-loaded processes did not distort fitting of photon distributions and calculation of FL. To clarify this, we have added these results and considerations to the manuscript (page 8, lines 13-16).

4. Relatedly, caution is warranted in interpreting variability within individual astrocytes. Two-photon resolution is limited in XYZ dimensions, and bolus loading may not reliably label fine processes. These limitations might temper the conclusions regarding process heterogeneity.

Our response: Thank you for this relevant comment. We kindly refer you to our response to comment 3.

5. Pharmacological perturbations: The authors examine the effects of various drugs. They find that TTX does not affect Na^+ levels, whereas gap junction blockade increases Na^+ , and NKA inhibition via low K^+ dramatically elevates Na^+ . These experiments would benefit from negative controls, such as using a dye or fluorescent protein insensitive to Na^+ .

Our response: Thank you for this suggestion. To identify any potential Na^+ -independent changes in FL introduced by the various drugs, we analyzed changes in the FL of the astrocyte marker SR101, present in all FLIM measurements, during the different protocols. We found that SR101-FL is independent from $[Na^+]$, as it did not change in low K^+ saline which resulted in large changes in ING-2 FL (reporting an increase in intracellular $[Na^+]$). We then reanalyzed experiments, in which we washed in TTX, CBX or (TFB-)TBOA and found that these drugs did not alter SR101-FL. To clarify this, we have included these results to the respective sections in the manuscript and prepared a new Supplementary Figure (Suppl. Fig. 5).

5a: In addition, clarification is needed on why TTX has no effect under basal conditions but does affect Na⁺ when levels are elevated by NKA blockade.

Our response: Thank you for this comment. When revisiting the results of this set of experiments, we noticed that the *p*-value in the previous Fig. 4i had been incorrectly indicated as “**” (two stars) when it should have been “*” (one star). We apologize for this error in the illustration of the data.

Indeed, we found that under basal conditions, perfusion of brain tissue slices with TTX did not result in detectable differences in the baseline [Na⁺]. In addition, the bimodal distribution of [Na⁺] remained. These results, now illustrated in new Figs. 4a and 4b, suggest that the different levels of baseline [Na⁺] in astrocytes are not primarily due to local spontaneous activity of microcircuits in brain tissue slices at rest. Moreover, it is worth noting here, that the release of glutamate activates Na⁺ import by EAATs, while the release of K⁺ activates NKA and Na⁺ export, thereby counteracting the influx of Na⁺ related to EAATs. It is therefore the sum of these opposing processes that determines the concentration of Na⁺ in astrocytes. These new points have been added to the results (page 9, lines 13-20).

To address the reviewer’s question concerning the effect of TTX upon removal of extracellular K⁺, we have performed additional electrophysiological recordings on CA1 pyramidal neurons. These confirmed the expected neuronal hyperpolarization upon perfusion with low K⁺. However, upon wash-in of standard extracellular K⁺, neurons rapidly repolarized and then underwent a prolonged phase of increased activity, which partly overlapped with the time window in which most astrocytes showed their peak increase in [Na⁺]. TTX prevented the phase of increased neuronal activity and action potential firing upon restoring standard extracellular K⁺. This new data is now described in the results (page 10, lines 13-16 and page 11, lines 1-4) and shown in the inset of the new Fig. 4k.

6. Response to K⁺ elevation: The authors report decreased Na⁺ in both somata and processes following high K⁺, with process Na⁺ showing sensitivity to TTX. This suggests compartment-specific regulation by synaptic activity. Since elevating K⁺ engages multiple pathways, complementary experiments using direct synaptic stimulation (electrical or pharmacological) would help isolate and strengthen the claims and mechanisms.

Our response: Thank you for this remark. Both our new data and our earlier studies fully support the reviewer's suggestion that astrocyte [Na⁺] is regulated in a compartment-specific manner by synaptic activity. Na⁺ uptake by EAATs is in fact a main mechanism responsible for transient increases in astrocytic [Na⁺]_s upon glutamatergic neuronal activity in physiological and pathophysiological conditions (Chatton *et al.*, 2016; Kirischuk *et al.*, 2016; Song *et al.*, 2020; Rose & Verkhratsky, 2024).

For example, using intensity-based widefield and two-photon Na⁺ imaging in astrocytes, we demonstrated earlier that local synaptic activity results in local Na⁺ transients in astrocyte processes (summarized e. g. in: Rose & Karus, 2013; Rose & Verkhratsky, 2024). Moreover, we could show that parallel fiber stimulation induces local Na⁺ transients in Bergmann glial processes, primarily due to glutamate transport (Bennay *et al.*, 2008). In astrocytes of the hippocampal CA1 *stratum radiatum*, brief afferent stimulation results in local Na⁺ transients, again mainly caused by activation of glutamate uptake (Langer & Rose, 2009). By contrast, synaptically-induced Na⁺ transients of neocortical astrocytes result from Na⁺ influx through both glutamate transporters and NMDA receptors (Ziemens *et al.*, 2019).

However, it is important to note that direct stimulation of glutamatergic synapses also activates several pathways, as synaptically-induced Na⁺ transients reflect an overlay of Na⁺ import through glutamate transport and concomitant Na⁺ export upon K⁺-induced activation of astrocytic

NKA as shown earlier (Karus *et al.*, 2015). In other words, local or global Na⁺ influx into astrocytes is counteracted by NKA-mediated Na⁺ efflux upon increases in [K⁺]_e (Karus *et al.*, 2015; Rose & Verkhratsky, 2024). We have clarified these points and added some of these ideas and additional information to the manuscript (pages 3, 9 and 19).

7. Modeling and NKA: The authors use simulations to explore how NKA configuration and expression affect Na⁺. However, the model treats astrocytes as single compartments, while their own data indicate somatic vs. process differences. They acknowledge this, but empirical validation would greatly strengthen the work. For example, testing how knockout/knockdown of specific NKA isoforms affects compartmental Na⁺ levels could directly support the model's predictions and enhance the rigor of their conclusions.

Our response: Thank you for this comment. Investigating the effect of manipulating specific NKA subunits on the Na⁺ concentration of astrocytes is a highly interesting suggestion, as it will most likely result in many new insights. At the same time, it is also very challenging in the context of the present study. Manipulating NKA subunits in astrocytes in intact tissue will almost certainly cause substantial changes to their functional properties and to the properties of neural networks, since NKA is one of the most fundamental plasma membrane transporters and is involved in many physiological processes. The first step in addressing this question would therefore be to knock-down specific NKA subunits in astrocytes in primary culture and determine the effect of this manipulation on their baseline Na⁺ as well as on changes in Na⁺ induced by e. g. lowering and increasing extracellular K⁺. Secondly, overexpressing the specific NKA subunits in cultured cells could help us to understand the previously observed effects. Finally, the effects of knocking out or overexpressing NKA subunits in brain tissue could be studied, followed by cell-type-specific manipulation of NKA. As mentioned above, these experiments are highly interesting, but are also expected to require substantial time, personnel and financial resources. Therefore, they certainly deserve and warrant a separate study.

In order to test the effect of a specific NKA composition on Na⁺ levels in different compartments (i.e. somata and processes), we extended the biophysical model. To simulate the spatial profile of [Na⁺] in an astrocyte, we extend the model such that it can simulate ion dynamics in a reconstructed morphology of an astrocyte (obtained from Neuromorpho.org (RRID:SCR_002145; (Kodali *et al.*, 2021); morphology #NMO_282188). Specifically, we varied the glutamate turnover rate of EAAT1 and the relative expression of NKA $\alpha 2\beta 1$ and $\alpha 2\beta 2$ isoforms as functions of distance from the soma to reproduce the observed spatial [Na⁺] profile. These new simulations show that the experimental observations in the spatial heterogeneity of [Na⁺] between soma and processes of individual astrocytes can be replicated by making the NKA expression and/or the Na⁺ influx through EAAT as functions of distance from the soma. The changes made to the model and the results from these new simulations are now incorporated in the revised manuscript (page 16, lines 13-24; new Fig. 8d and Supplementary Methods Text 2).

Minor point: Several parameter values are reported in text but not shown as data in the figures. I don't see any reason not to show these data in some format in figures or supplementary figures.

Our response: As suggested, we have prepared a new Supplementary Figure (new Suppl. Fig. 6) to illustrate data not shown in our original manuscript.

In addition to these revisions, we have made minor textual changes to the manuscript to remove remaining errors and typos. Moreover, we had to shorten the original manuscript text throughout in

order to comply with the required word limit after incorporating the additional data and information suggested by the reviewers. This required rewriting some sentences and sections.

References listed in the rebuttal letter:

- Baranauskas G, David Y & Fleidervish IA (2013). Spatial mismatch between the Na⁺ flux and spike initiation in axon initial segment. *Proc Natl Acad Sci U S A* **110**, 4051. doi:10.1073/pnas.1215125110
- Bennay M, Langer J, Meier SD, Kafitz KW & Rose CR (2008). Sodium signals in cerebellar Purkinje neurons and Bergmann glial cells evoked by glutamatergic synaptic transmission. *Glia* **56**, 1138. doi:10.1002/glia.20685
- Bindocci E, Savtchouk I, Liaudet N, Becker D, Carriero G & Volterra A (2017). Three-dimensional Ca²⁺ imaging advances understanding of astrocyte biology. *Science* **356**. doi:10.1126/science.aai8185
- Chatton JY, Magistretti PJ & Barros LF (2016). Sodium signaling and astrocyte energy metabolism. *Glia* **64**, 1667. doi:10.1002/glia.22971
- Ewald AJ, Werb Z & Egeblad M (2011). Monitoring of vital signs for long-term survival of mice under anesthesia. *Cold Spring Harbor protocols* **2011**, pdb prot5563. doi:10.1101/pdb.prot5563
- Fleidervish IA, Lasser-Ross N, Gutnick MJ & Ross WN (2010). Na⁺ imaging reveals little difference in action potential-evoked Na⁺ influx between axon and soma. *Nat Neurosci* **13**, 852. doi:10.1038/nn.2574
- Hanson E, Armbruster M, Cantu D, Andresen L, Taylor A, Danbolt NC & Dulla CG (2015). Astrocytic glutamate uptake is slow and does not limit neuronal NMDA receptor activation in the neonatal neocortex. *Glia* **63**, 1784. doi:10.1002/glia.22844
- Jabaudon D, Shimamoto K, Yasuda-Kamatani Y, Scanziani M, Gahwiler BH & Gerber U (1999). Inhibition of uptake unmasks rapid extracellular turnover of glutamate of nonvesicular origin. *Proc Natl Acad Sci U S A* **96**, 8733.
- Kafitz KW, Meier SD, Stephan J & Rose CR (2008). Developmental profile and properties of sulforhodamine 101-labeled glial cells in acute brain slices of rat hippocampus. *J Neurosci Methods* **169**, 84. doi:10.1016/j.jneumeth.2007.11.022
- Karpf J, Unichenko P, Chalmers N, Beyer F, Wittmann MT, Schneider J, Fidan E, Reis A, Beckervordersandforth J, Brandner S, Liebner S, Falk S, Sagner A, Henneberger C & Beckervordersandforth R (2022). Dentate gyrus astrocytes exhibit layer-specific molecular, morphological and physiological features. *Nat Neurosci* **25**, 1626. doi:10.1038/s41593-022-01192-5
- Karus C, Mondragao MA, Ziemens D & Rose CR (2015). Astrocytes restrict discharge duration and neuronal sodium loads during recurrent network activity. *Glia* **63**, 936. doi:10.1002/glia.22793
- King CM, Bohmbach K, Minge D, Delekate A, Zheng K, Reynolds J, Rakers C, Zeug A, Petzold GC, Rusakov DA & Henneberger C (2020). Local Resting Ca(2+) Controls the Scale of Astroglial Ca(2+) Signals. *Cell reports* **30**, 3466. doi:10.1016/j.celrep.2020.02.043
- Kirischuk S, Heja L, Kardos J & Billups B (2016). Astrocyte sodium signaling and the regulation of neurotransmission. *Glia* **64**, 1655. doi:10.1002/glia.22943
- Kodali M, Attaluri S, Madhu LN, Shuai B, Upadhyaya R, Gonzalez JJ, Rao X & Shetty AK (2021). Metformin treatment in late middle age improves cognitive function with alleviation of

- microglial activation and enhancement of autophagy in the hippocampus. *Aging Cell* **20**, e13277. doi:10.1111/accel.13277
- Kotler O, Khrapunsky Y & Fleidervish I (2023). Measuring Action Potential Propagation Velocity in Murine Cortical Axons. *Bio Protoc* **13**, e4876. doi:10.21769/BioProtoc.4876
- Langer J, Gerkau NJ, Derouiche A, Kleinhans C, Moshrefi-Ravasdjani B, Fredrich M, Kafitz KW, Seifert G, Steinhauser C & Rose CR (2017). Rapid sodium signaling couples glutamate uptake to breakdown of ATP in perivascular astrocyte endfeet. *Glia* **65**, 293. doi:10.1002/glia.23092
- Langer J & Rose CR (2009). Synaptically induced sodium signals in hippocampal astrocytes in situ. *J Physiol* **587**, 5859. doi:10.1113/jphysiol.2009.182279
- Langer J, Stephan J, Theis M & Rose CR (2012). Gap junctions mediate intercellular spread of sodium between hippocampal astrocytes in situ. *Glia* **60**, 239. doi:10.1002/glia.21259
- Larsen BR, Stoica A & MacAulay N (2019). Developmental maturation of activity-induced K(+) and pH transients and the associated extracellular space dynamics in the rat hippocampus. *J Physiol* **597**, 583. doi:10.1113/JP276768
- Meyer J, Gerkau NJ, Kafitz KW, Patting M, Jolmes F, Henneberger C & Rose CR (2022). Rapid fluorescence lifetime imaging reveals that TRPV4 channels promote dysregulation of neuronal Na(+) in ischemia. *J Neurosci* **42**, 552. doi:10.1523/JNEUROSCI.0819-21.2021
- Mondragao MA, Schmidt H, Kleinhans C, Langer J, Kafitz KW & Rose CR (2016). Extrusion versus diffusion: mechanisms for recovery from sodium loads in mouse CA1 pyramidal neurons. *J Physiol* **594**, 5507. doi:10.1113/JP272431
- Nelson JSE, Meyer J, Gerkau NJ, Kafitz KW, Ullah G, Santamaria F & Rose CR (2025). Spatio-temporal dynamics of lateral Na(+) diffusion in apical dendrites of mouse CA1 pyramidal neurons. *J Neurosci* **45**, e0077252025. doi:10.1523/JNEUROSCI.0077-25.2025
- Pietrobon D & Conti F (2024). Astrocytic Na(+), K(+) ATPases in physiology and pathophysiology. *Cell Calcium* **118**, 102851. doi:10.1016/j.ceca.2024.102851
- Rose CR & Karus C (2013). Two sides of the same coin: sodium homeostasis and signaling in astrocytes under physiological and pathophysiological conditions. *Glia* **61**, 1191. doi:10.1002/glia.22492
- Rose CR, Kovalchuk Y, Eilers J & Konnerth A (1999). Two-photon Na⁺ imaging in spines and fine dendrites of central neurons. *Pflugers Arch* **439**, 201.
- Rose CR & Ransom BR (1997). Gap junctions equalize intracellular Na⁺ concentration in astrocytes. *Glia* **20**, 299.
- Rose CR & Verkhratsky A (2024). Sodium homeostasis and signalling: The core and the hub of astrocyte function. *Cell Calcium* **117**, 102817. doi:10.1016/j.ceca.2023.102817
- Rothstein JD, Dykes-Hoberg M, Pardo CA, Bristol LA, Jin L, Kuncl RW, Kanai Y, Hediger MA, Wang Y, Schielke JP & Welty DF (1996). Knockout of glutamate transporters reveals a major role for astroglial transport in excitotoxicity and clearance of glutamate. *Neuron* **16**, 675.
- Schreiner AE, Durry S, Aida T, Stock MC, Rütger U, Tanaka K, Rose CR & Kafitz KW (2014). Laminar and subcellular heterogeneity of GLAST and GLT-1 immunoreactivity in the developing postnatal mouse hippocampus. *Journal of Comparative Neurology* **522**, 204. doi:10.1002/cne.23450
- Semyanov A, Henneberger C & Agarwal A (2020). Making sense of astrocytic calcium signals - from acquisition to interpretation. *Nat Rev Neurosci* **21**, 551. doi:10.1038/s41583-020-0361-8
- Shigetomi E, Patel S & Khakh BS (2016). Probing the Complexities of Astrocyte Calcium Signaling. *Trends Cell Biol* **26**, 300. doi:10.1016/j.tcb.2016.01.003
- Skou JC & Esmann M (1992). The Na,K-ATPase. *J Bioenerg Biomembr* **24**, 249.

- Song S, Luo L, Sun B & Sun D (2020). Roles of glial ion transporters in brain diseases. *Glia* **68**, 472. doi:10.1002/glia.23699
- Thrane AS, Thrane VR, Zeppenfeld D, Lou N, Xu Q, Nagelhus EA & Nedergaard M (2012). General anesthesia selectively disrupts astrocyte calcium signaling in the awake mouse cortex. *Proc Natl Acad Sci U S A* **109**, 18974. doi:10.1073/pnas.1209448109
- Volterra A, Liaudet N & Savtchouk I (2014). Astrocyte Ca(2+) signalling: an unexpected complexity. *Nat Rev Neurosci* **15**, 327. doi:10.1038/nrn3725
- Weilinger NL, Wicki-Stordeur LE, Groten CJ, LeDue JM, Kahle KT & MacVicar BA (2022). KCC2 drives chloride microdomain formation in dendritic blebbing. *Cell reports* **41**, 111556. doi:10.1016/j.celrep.2022.111556
- Zheng K, Bard L, Reynolds JP, King C, Jensen TP, Gourine AV & Rusakov DA (2015). Time-Resolved Imaging Reveals Heterogeneous Landscapes of Nanomolar Ca(2+) in Neurons and Astroglia. *Neuron* **88**, 277. doi:10.1016/j.neuron.2015.09.043
- Ziemens D, Oschmann F, Gerkau NJ & Rose CR (2019). Heterogeneity of activity-induced sodium transients between astrocytes of the mouse hippocampus and neocortex: Mechanisms and consequences. *J Neurosci* **39**, 2620. doi:10.1523/JNEUROSCI.2029-18.2019